

# Wildfire smoke triggers cirrus formation: Lidar observations over the Eastern Mediterranean (Cyprus)

Rodanthi-Elisavet Mamouri[1,4], Albert Ansmann[2], Kevin Ohneiser[2], Daniel A. Knopf[3],
Argyro Nisantzi[1,4], Johannes Bühl[5], Ronny Engelmann[2], Annett Skupin[2], Patric Seifert[2], Holger Baars[2],
Dragos Ene[1], Ulla Wandinger[2], and Diofantos Hadjimitsis[1,4]

[1]Eratosthenes Centre of Excellence, Limassol, Cyprus
[2]Leibniz Institute for Tropospheric Research, Leipzig, Germany
[3]School of Marine and Atmospheric Sciences, Stony Brook University, Stony Brook, NY 11794-5000, USA
[4]Cyprus University of Technology, Dep. of Civil Engineering and Geomatics, Limassol, Cyprus
[5]Technical University Wernigerode, Wernigerode, Germany

**Correspondence:** R. E. Mamouri (rodanthi.mamouri@eratosthenes.org.cy), A. Ansmann (albert@tropos.de)

**Abstract.**

The number of intense wildfires may increase in the upcoming years as a consequence of climate change. Changing aerosol conditions may lead to changes in regional and global cloud and precipitation pattern. One key aspect of research is presently whether or not wildfire smoke particles can initiate ice nucleation. We found strong evidence that aged smoke particles (dominated by organic aerosol particles) originating from wildfires in North America triggered significant ice nucleation at temperatures from$-47$ to $-53°$C and caused the formation of extended cirrus layers. Our study is based on lidar observations over Limassol, Cyprus, from 27 October to 3 November 2020 when extended wildfire smoke fields crossed the Mediterranean Basin from Portugal to Cyprus. The observations suggest that the ice crystals were nucleated just below the tropopause in the presence of smoke particles serving as ice-nucleating particles (INPs). The main part of the 2-3 km thick smoke layer was, however, in the lower stratosphere just above the tropopause. With actual radiosonde observations of temperature and relative humidity and lidar-derived smoke particle surface area concentrations as starting values, gravity wave simulations show that lofting by 90-180 m is sufficient to initiate significant ice nucleation on the smoke particles, expressed in ice crystal number concentrations of 1-100 L$^{-1}$.

## 1 Introduction

Record-breaking wildfires in western Canada (2017), southeastern Australia (2019-2020), and central Siberia (2019) caused strong perturbations of the aerosol conditions in the upper troposphere and lower stratosphere (UTLS) (Baars et al., 2019; Kloss et al., 2019; Ohneiser et al., 2021, 2022; Rieger et al., 2021). The smoke was lofted by pyrocumulonimbus (pyroCb) convection (Peterson et al., 2018, 2021) or self-lofting processes (Ohneiser et al., 2021, 2023) into the UTLS height range, traveled around the globe, and polluted large parts of the northern and southern hemisphere from the subtropics to the poles over months. Recent studies suggest that major fire-related hemispheric perturbations may become more frequent in future within a changing global climate with more hot and dry weather periods (Jolly et al., 2015; Abatzoglou et al., 2019; Kirchmeier-Young et al., 2019).



Besides extraordinarily intense fires storms, numerous small to moderate wildfires and biomass burning events all over the world serve as a persistent source for smoke particles in the free troposphere (Mattis et al., 2008; Dahlkötter et al., 2014; Burton et al., 2015; Vaughan et al., 2018; Baars et al., 2021; Foth et al., 2019; Floutsi et al., 2021; Hu et al., 2022; Veselovskii et al., 2022; Michailidis et al., 2023). Based on airborne pole-to-pole in situ aerosol observations during the Atmospheric Tomography (ATom) mission, Schill et al. (2020) found biomass burning particles (organic aerosol, OA) and sulfate as the major contributors by mass to submicron aerosols in the remote free troposphere.

To be able to adequately consider fresh and aged wildfire smoke in global chemistry and climate models (Hodzic et al., 2020), the role of smoke particles in the atmospheric system needs to be explored in field studies (in situ, remote sensing), in-depth laboratory experiments, and by means of cloud-resolving atmospheric modeling on regional to global scales. It was already shown that smoke particles in the stratosphere can significantly influence climate conditions (i.e., radiation, dynamics) (Das et al., 2021; Hirsch and Koren, 2021; Stocker et al., 2021; Yu et al., 2021; Heinold et al., 2022; Sellitto et al., 2022; Damany-Pearce et al., 2022; Senf et al., 2023) and stratospheric ozone depletion (Ohneiser et al., 2021, 2022; Rieger et al., 2021; Ansmann et al., 2022; Yook et al., 2022; Solomon et al., 2022, 2023). In the upper troposphere, smoke particles are suggested to be able to serve as ice-nucleating particles (INPs) in cirrus formation processes (Cziczo et al., 2013; Knopf et al., 2018; Wolf et al., 2020; Ansmann et al., 2021; Raga et al., 2022). First observations of smoke-cirrus interaction in the Arctic support this hypothesis (Engelmann et al., 2021; Ansmann et al., 2023). This article will provide further evidence that smoke particles have the potential to initiate ice formation in the upper troposphere.

Because of the complex chemical, microphysical, and morphological properties of aged fire smoke particles, which can occur as glassy, semi-solid, and liquid aerosol particles, the development of smoke INP parameterization schemes (to be used in models) is a crucial task (Knopf et al., 2018; Knopf and Alpert, 2023). The particles and released vapors in young biomass burning plumes undergo chemical and physical aging processes on their way up to the tropopause and during long-range transport over weeks and months. Aging includes photo-chemical processes, heterogeneous chemical reactions on and in the particles, condensation of gases on the particle surfaces, collision and coagulation, and cloud processing (when acting as cloud condensation nuclei, CCN, or INPs in several consecutive cloud evolution and dissipation events). All these impacts change the chemical composition of the smoke particles, their morphological characteristics (size, shape, and internal structure), their internal mixing state, and thus their ice nucleation efficiency.

It is assumed that smoke particles, after finalizing the aging process, show an almost perfect spherical core-shell structure with a black-carbon-containing core and an organic-material-rich shell, and that the ability to serve as INP mainly depends on the material in the particle's shell (Charnawskas et al., 2017). Several studies indicate that aged smoke particle from forest fires contain only 2-3% black carbon (BC) (Dahlkötter et al., 2014; Yu et al., 2019; Torres et al., 2020; Ohneiser et al., 2023) so that the organic substances (organic carbon, OC) are responsible for the ice nucleation activity. Biomass-burning particles also contain humic like substances (HULIS) which represent large macromolecules that may serve as INP at low temperatures of $-50$ to $-70°C$ (Wang and Knopf, 2011; Knopf et al., 2018). If the particles are in a glassy state, they can serve as deposition ice nucleation (DIN) INPs (Zobrist et al., 2008; Murray et al., 2010; Wang et al., 2012; Berkemeier et al., 2014; Knopf et al., 2018). DIN is defined as ice formation occurring on the INP surface by water vapor deposition from the supersaturated gas





phase. DIN could be th result of pore condensation freezing (Marcolli, 2014; Knopf and Alpert, 2023). When the supercooled smoke particles take up water and their shell deliquesces, immersion freezing can proceed, where the remaining solid part of the particle (immersed in the liquid shell) serves as INP. If the smoke particles become completely liquid (and no insoluble
material within the particles is left), homogeneous freezing will take place at temperatures below −38°C (Knopf and Alpert, 2023).

It was shown that wildfire smoke particles are inefficient INPs in mixed-phase cloud processes with temperatures of −30°C and higher (Barry et al., 2021). Similarly, Froyd et al. (2010) found no indication that wildfire smoke particles influenced cirrus formation. However, they found an aerosol mixture containing dust, and when dust is present, it will dominate ice nucleation
so that ice supersaturation thresholds needed to activate smoke particles may not be reached. Jahn et al. (2020) and Jahl et al. (2021) hypothesized that aged smoke particles contain minerals and that these components determine the smoke INP efficacy. It remains to be shown whether such smoke particles are able to influence mixed-phase and cirrus cloud developments. Those INPs should already be ice-active at temperatures of −20 to −30°C. Hence, in regions of convection, the upper troposphere may be depleted of the most active smoke INPs.

In contrast to the airborne studies on smoke-cirrus interactions, we found clear evidence for an impact of aged wildfire smoke on cirrus formation during the MOSAiC (Multidisciplinary drifting Observatory for the Study of Arctic Climate) expedition (Engelmann et al., 2021; Ansmann et al., 2023). Work is in progress to systematically analyze a large number of cirrus systems that developed in wildfire-smoke polluted air over the central Arctic during the MOSAiC winter halfyear of 2019-2020 and were observed with lidar and radar aboard the German icebreaker Polarstern.

In this article here, we will discuss a series of lidar observations of ice clouds that were generated in aged North American wildfire smoke layers at the tropopause. The measurements were performed at Limassol, Cyprus, in the Eastern Mediterranean in October-November 2020. Ice nucleation started in the lowest part of these smoke layers (just below the tropopause). The freshly nucleated ice crystals formed extended fields of long fall streaks (virga). The simultaneous occurrence of smoke layers together with intense cirrus features is a strong sign that smoke particles were serving as the dominant INPs.

The article is organized as follows: In Sect. 2, we provide information about the Limassol remote sensing station. The lidar instrument and data products are briefly described in Sect. 3. The smoke INP parameterization is outlined in Sect. 4. Observations of the smoke layer, the smoke optical properties, and cirrus layers that formed in the smoke are then presented in Sect. 5. A short summary is given in Sect. 6.

## 2 Cyprus Atmospheric Remote Sensing Observatory (CARO)

A wind Doppler lidar and the multiwavelength polarization Raman lidar Polly (POrtabLe Lidar sYstem) (Engelmann et al., 2016) is presently operated at the Cyprus Atmospheric Remote Sensing Observatory (CARO) of the Eratosthenes Centre of Excellence at Limassol (34.677°N, 33.0375°E, 2.8 m above sea level, a.s.l.) for continuous vertical profiling of the wind field and aerosol conditions. The observatory will be additionally equipped, in 2024, with a containerized cloud radar, microwave radiometer, and a disdrometer for cloud and precipitation monitoring. One of the main research topics comprises the interaction



between aerosols, clouds, precipitation, and atmospheric dynamics in the highly polluted Eastern Mediterranean where complex mixtures of desert and soil dust, biogenic particle components, and anthropogenic haze regularly occur (Nisantzi et al., 2014; Rogozovsky et al., 2021; Heese et al., 2022). CARO is part of the ACTRIS (Aerosols, Clouds and Trace gases Research InfraStructure) National Facility of the Republic of Cyprus for remote sensing of aerosols and clouds (ACTRIS, 2023). The Limassol lidar station is part of PollyNET (PollyNET, 2023), a network of continuously operated Polly lidar stations (Baars et al., 2016) and of the European Aerosol Research Lidar Network (EARLINET) (Pappalardo et al., 2014) organized within the ACTRIS project. A sunphotometer is operated in addition at Limassol since 2010 in the framework of AERONET (Aerosol Robotic Network, CUT-TEPAK station) (Holben et al., 1998; AERONET, 2023). Combined lidar and photometer observations are performed at Cyprus mainly with focus on dust research since 2012 (Mamouri et al., 2013, 2016; Nisantzi et al., 2015; Ansmann et al., 2019a). Meanwhile also smoke is a topic of research (Nisantzi et al., 2014; Baars et al., 2019).

## 3 Polly instrument and primary data analysis

The setup and basic technical details of the Polly instrument are given in Engelmann et al. (2016), Hofer et al. (2017), and Jimenez et al. (2020). The diode-pumped laser transmits linearly polarized laser pulses at 355, 532, and 1064 nm with a pulse repetition rate of 100 Hz. All laser beams are pointing to an off-zenith angle of 5° to avoid a bias in the observations of the optical properties of mixed-phase and cirrus clouds caused by strong specular reflection by falling and then frequently horizontally aligned ice crystals.

A detailed description of the Polly data analysis regarding particle optical properties can be found in Baars et al. (2016), Hofer et al. (2017), Jimenez et al. (2020), and Ohneiser et al. (2020, 2021, 2022). The polarization Raman lidar permits us to measure height profiles of the particle backscatter coefficient $\beta$ at the laser wavelengths of 355, 532 and 1064 nm, particle extinction coefficient $\alpha$ at 355 and 532 nm, the corresponding lidar ratios $L = \alpha/\beta$, and the volume and particle linear depolarization ratios at 355 and 532 nm. Uncertainties in the lidar products are about 5-10% (particle backscatter coefficient, linear depolarization ratio), 10-25% (particle extinction coefficient), and 15-30% (extinction-to-backscatter ratio).

We will especially make use of the height profiles of the 532 nm particle backscatter coefficients at smoke and cirrus height level. The so-called Klett-Fernald method (Klett, 1981; Fernald, 1984) allows us to determine profiles of the backscatter coefficient from strong elastic-backscatter signal profiles. That means the profiles can be obtained with high vertical and temporal resolution. However, the lidar ratio must by given as input and may cause large uncertainties (20-40% relative uncertainty). In contrast, the Raman lidar method (Ansmann et al., 1992) does not need critical input parameters, and thus is more accurate. However, this method is based on the analysis of height profiles of the ratio of the elastic-backscatter signal to the respective Raman signal. To keep the influence of enhanced signal noise low, longer vertical smoothing and longer signal averaging times are required when using this method.



## 4   POLIPHON and INP parameterization

The POLIPHON (POlarization LIdar PHOtometer Networking) method (Mamouri and Ansmann, 2016, 2017) enables us to retrieve aerosol-type-dependent microphysical products from the measured height profiles of the particle backscatter coefficient and to estimate cloud-process-relevant properties such as CCN and INP concentrations. A detailed view on the POLIPHON potential regarding dust and wildfire smoke retrievals is given by Ansmann et al. (2019b, 2021).

In this study, we make use of the conversion of 532 nm backscatter coefficients into particle surface area concentration $s$ and particle number concentration $n_{250}$ (number concentration of particles with radius >250 nm). $s$ is the smoke input parameter in the INP parameterization, described in the next section, and $n_{250}$ can be regarded as a rough proxy for the INP reservoir (available particles that could potentially be activated as INP).

The following relationships are used to calculate $s$ and $n_{250}$ (Mamouri and Ansmann, 2017; Ansmann et al., 2021):

$$s(z) = c_s L \beta(z), \qquad (1)$$

$$n_{250}(z) = c_{250} L \beta(z) \qquad (2)$$

with the 532 nm particle backscatter coefficient $\beta(z)$ at height $z$ and the extinction-to-backscatter or lidar ratio $L$. For the wildfire smoke episode in October 2020, we measured smoke lidar ratios of around 75 sr as shown in Sect. 5.1. Lidar ratios of >70 sr are indicative for strongly light-absorbing smoke particles (Haarig et al., 2018; Ohneiser et al., 2020, 2022). The extinction-to-surface-area conversion factor $c_s$ and the extinction-to-number conversion factor $c_{250}$ for 532 nm are 1.75 Mm $\mu m^2$ cm$^{-3}$ and 0.35 Mm cm$^{-3}$, respectively (Ansmann et al., 2021). Considering an uncertainty of 25% in the conversion factors and a lidar ratio uncertainty of about 15-20%, we can obtain the microphysical properties with a relative uncertainty of about 30%.

These conversion factors for aged wildfire smoke are determined at dry aerosol conditions (Ansmann et al., 2021) and thus hold for dry aerosol particles. Since the $s$ and $n_{250}$ retrievals are based on lidar observations performed during cirrus free conditions, a potential bias (overestimation of dry particle backscatter and thus of $s$ and $n_{250}$) caused by water uptake by the smoke particles in the upper, usually, dry troposphere is neglected here.

### 4.1   INP parameterization

The estimation of INP concentrations is challenging due to the chemical complexity of smoke aerosol (Kanji et al., 2017; Knopf et al., 2018; Jahn et al., 2020). As mentioned in the introduction, we assume that aged smoke particles show a core-shell structure with a BC-containing core and an OC-rich shell, and that the ability to serve as INP mainly depends on the material in the shell and thus on the organic material of the particles and on the thermodynamic state (glassy, semi-solid, liquid) (Berkemeier et al., 2014; Knopf et al., 2018). In the following, we briefly outline procedures to compute INP concentrations for immersion freezing and deposition ice nucleation.

Knopf and Alpert (2013) introduced the water-activity-based immersion freezing model (ABIFM) and presented the respective parameterization for two types of humic compounds based on experimental data (Rigg et al., 2013). We chose to apply the ABIFM for Leonardite (a standard humic acid surrogate material) to represent the amorphous organic coating of smoke



particles. The ABIFM allows the prediction of the ice particle production rate $J_{\text{het}}^{\text{IF}}$ as a function of ambient air temperature $T$ (freezing temperature) and ice supersaturation $S_{\text{ICE}}$ (Knopf and Alpert, 2023). In the first step, the so-called water activity criterion is computed (Koop et al., 2000):

$$\Delta a_{\text{w}} = a_{\text{w}} - a_{\text{w,i}}(T) \tag{3}$$

with the decimal value of relative humidity (RH) equals condensed-phase water activity, $a_{\text{w}}$, when particle and ambient RH are in equilibrium. $a_{\text{w,i}}$ denotes the water activity at the ice melting point for a given aqueous solution and is expressed as the ratio of the ice saturation pressure $p_{\text{i}}$ to water saturation pressure $p_{\text{w}}$ as function of temperature $T$ in the air parcel in which ice nucleation takes place. RH and temperature values may be available from radiosonde ascents or taken from data bases with re-analyzed global atmospheric data. In the next step, the ice crystal nucleation rate coefficient $J_{\text{het}}^{\text{IF}}$ (in $\text{cm}^{-2}\,\text{s}^{-1}$) is calculated:

$$\log_{10}(J_{\text{het}}^{\text{IF}}) = b + k\Delta a_{\text{w}}. \tag{4}$$

The particle parameters $b$ and $k$ are determined from laboratory studies. For Leonardite, $b$ is $-13.4$ and $k$ is $66.9$ (Knopf and Alpert, 2013). In the final step, we obtain the number concentration of smoke INP for the immersion freezing mode,

$$n_{\text{INP}}^{\text{IF}} = s J_{\text{het}}^{\text{IF}} \Delta t \tag{5}$$

with the particle surface area concentration $s$ and the time period $\Delta t$ (in seconds) during which $T$ and $S_{\text{ICE}}$ are constant. In our gravity wave simulation in Sect. 5.3, the time step is $\Delta t = 20$ s (in simulations of waves with periods of 1200 s).

Wang and Knopf (2011) provide a simplified parameterization of DIN, based on classical nucleation theory, that describes the DIN efficiency and the ice crystal nucleation rate $J_{\text{het}}^{\text{DIN}}$ as a function of ambient temperature $T$, $S_{\text{ICE}}$, and the ice-nucleation-relevant aerosol composition, e.g., humic and fulvic acid compounds. The procedure (for lidar applications) is outlined in Ansmann et al. (2021). The final step is then also here to compute the DIN INP concentration which expresses the predicted ice crystal number concentration:

$$n_{\text{INP}}^{\text{DIN}} = s J_{\text{het}}^{\text{DIN}} \Delta t. \tag{6}$$

### 4.2 Nicosia radiosonde

The Athalassa Radiosonde Station 17607 (35.14°N, 33.39°E, 160 m a.s.l.), Nicosia, Cyprus launches two Vaisala RS41-SGP radiosondes daily, at 05:00 and 11:00 UTC. The sonde measures pressure, temperature, relative humidity, horizontal wind velocity and direction (Nicosia-Athalassa-RS, 2023). The radiosonde station is 61.4 km northeast of CARO.

## 5 Observations

### 5.1 Smoke identification and optical characterization

From 21 October to 3 November 2020 extended North American wildfire smoke layers crossed the Mediterranean Basin from Portugal to Cyprus between 6 and 14 km height (Michailidis et al., 2023). Lidar measurements at Evora, Portugal, and Potenza,





Italy, documented this event on 24 and 26 October 2020, respectively. The smoke originated from large wildfires in California, USA, and traveled 8 days before reaching Europe. Figure 1 shows a smoke measurement with the Polly instrument at Cyprus on 27 October 2020 (PollyNET, 2023). Weak aerosol structures are visible from 6-11 km height, and pronounced layers were

detected from 11-14 km height. In Fig. 2, backward trajectories confirm the 8-9 day long-range transport of smoke from the western United States to the Eastern Mediterranean.

The strong wavelength dependence of the backscatter coefficient and the respectively high backscatter Ångström exponents of 1-2, and the weak wavelength dependence of the extinction coefficient are typical for aged, strongly light-absorbing wildfire smoke. In the optically thickest part from 12-13 km height with highest particle extinction coefficients, the lidar ratio

was about 70-90 sr at 532 nm and 50-60 sr at 355 nm. This inverse spectral dependence of the lidar ratio is characteristic for aged wildfire smoke. The main smoke layer was above the tropopause.

The enhanced particle depolarization ratios of 0.1-0.15 at both wavelengths indicate non-spherical particles. The particles were probably lofted by pyroCb convection to the tropopause region. The fast lofting into the dry upper troposphere is assumed to prohibit aging of the particles, i.e., condensation of gases emitted over the fire places as well as of water vapor on the freshly

emitted irregularly shaped particles, and thus the development of a spherical core-shell structure (causing low depolarization ratios close to zero) (Haarig et al., 2018; Ohneiser et al., 2020; Ansmann et al., 2021). Lofting of smoke by pyroCbs up to the tropopause (at 13-14 km over California in October 2020) is, however, not considered in the HYSPLIT (Hybrid Single-Particle Lagrangian Integrated Trajectory model) simulations (Stein et al., 2015; Rolph et al., 2017). Thus, the trajectories can only be used to follow the smoke-polluted airmasses on their way towards Europe. Our analyses are in good agreement with the lidar

measurements at Evora, Portugal, and Potenza, Italy, and HYSPLIT backward trajetory analysis presented by Michailidis et al. (2023).

## 5.2 Ice nucleation in wildfire smoke

On 28 and 30 October 2020, ice clouds developed at the tropopause over Cyprus. According to Figs. 3b and d, the main smoke layer was located between the tropopause and 12-12.5 km height on these two days. The sharp drop in the RH profiles in

Figs. 3a and c at 10.5 km (28 October) and 11 km (30 October) indicates the tropopause. We show the volume depolarization ratio in Fig. 3b to better identify the smoke layers at 10-12 km and at 6 km height on this day. The white, tilted column-like features in Figs. 3b and d are ice virga consisting of falling ice crystals. The nucleation of ice crystals on smoke particles most probably started at the top of the humid layer in the coldest part of the troposphere (at temperatures from $-47$ to $-53$°C). These ice crystals grew fast in the supersaturated air and immediately started to fall. Well-structured coherent virga are formed

by these crystals. The hexagonal ice crystals cause strong depolarization ratios around 40%. The virga are visible as long as the RH is high so that sublimation of ice crystals, even in subsaturated air, is slow or prohibited.

Cirrus formation intensified on 30 October from 6 UTC to 12 UTC and the optical depth of the virga increased so that the smoke layer above the virga was no longer visible in Fig. 3d (after 9:45 UTC). The color plot is based on lidar profiles measured with 7.5 m vertical and 30 s temporal resolution. Averaging of signal profiles over, e.g., 15-20 minutes and vertical





smoothing with window lengths of 150-750 m is required to resolve the full cirrus and smoke layer structures up to the smoke layer top.

Strong ice nucleation and virga evolution were observed over many hours in the evening of 30 October 2020. Figs. 3 and 4 clearly suggest that the smoke particles served as INPs (at temperatures around $-50$°C). The top height of the virga zone always coincided with the lower part of the smoke layer. By using the classical Raman lidar technique (Ansmann et al.,
1992; Wandinger, 1998) we analyzed the cirrus virga on 30 October 2020 in detail. The multiple-scattering-corrected 532 nm extinction coefficient $\sigma_{532}$ of the ice crystals reached values as high as 400 $\mathrm{Mm}^{-1}$ (10:00-10:30 UTC mean values for the 8-11 km height range in Fig. 3d), 180 $\mathrm{Mm}^{-1}$ (19:40-21:20 UTC, 8-10 km height range in Fig. 4), and 1000 $\mathrm{Mm}^{-1}$ (23:40-24:00 UTC, 8-10 km height range in Fig. 4). The respective cirrus optical depths were 0.8 (10:00-10:30 UTC), 0.25-0.3 (19:40-21.20 UTC), and 1.8-2.0 (23:40-24:00 UTC).

The pronounced virga structures point to a relatively small number of comparably large ice crystals that grew fast and formed these well-organized virga signatures. A broad crystal size spectrum (causing a respectively broad spectrum of sedimentation velocities) would probably not be able to produce such coherent virga structures over many hours. Patchy and incoherent cirrus structures would be more likely. We estimated the ice crystal number concentration $N_{\mathrm{ICE}}$ by using the relationship of $\sigma_{532} \approx N_{\mathrm{ICE}} 2\pi r_{\mathrm{eff}}^2$ (Schumann et al., 2011) with the measured 532 nm extinction coefficient $\sigma_{532}$ and values for the assumed
effective crystal radius $r_{\mathrm{eff}}$. For $\sigma_{532} = 400\,\mathrm{Mm}^{-1}$ (as observed on 30 October, 10:00-10:30 UTC), we yield $N_{\mathrm{ICE}} = 100\,\mathrm{L}^{-1}$ ($r_{\mathrm{eff}} = 25\,\mu\mathrm{m}$), 25 $\mathrm{L}^{-1}$ ($r_{\mathrm{eff}} = 50\,\mu\mathrm{m}$), and 6 $\mathrm{L}^{-1}$ ($r_{\mathrm{eff}} = 100\,\mu\mathrm{m}$). These estimated values of $N_{\mathrm{ICE}}$ will be compared with estimated (simulated) INP concentrations in Sect. 5.3.

Figures 5 and 6 provide a more detailed, quantitative insight into the impact of smoke on ice nucleation. As can be seen, the smoke layer sufficiently overlapped with the humid region in the uppermost troposphere and thus was able to influence
the development of cirrus clouds and virga significantly. In Fig. 5, we used both methods (Klett-Fernald approach, Raman lidar approach, see Sect. 3) to compute smoke backscatter profiles. Then, we multiplied the backscatter coefficients with the smoke lidar ratio of 75 sr to obtain the respective particle extinction coefficients for several cirrus-free time periods on 28 and 30 October 2020. Very polluted conditions prevailed. High particle extinction coefficients of up to 250 $\mathrm{Mm}^{-1}$ (28 October) and 150 $\mathrm{Mm}^{-1}$ (30 October) were observed. The aerosol optical thickness (AOT, 532 nm) of the smoke layer (8-12.5 km height
range) was close to 0.2 on 28 October and about 0.16 on 30 October (9.5-12.5 km height range). Good agreement with the early morning AERONET photometer observations at Limassol, Cyprus, of the total 500 nm AOT was found (AERONET, 2023). Besides the smoke, dust and urban haze (below 5 km height in Fig. 3) contributed to the overall 500 nm AOT.

The lidar observations in Fig. 5 show the aerosol conditions before first ice clouds and virga developed on these days. In Fig. 6, particle surface area concentrations (PSAC) are presented, obtained by conversion of the extinction coefficients in Fig. 5
into surface area values by means of Eq. (1) in Sect. 4. In terms of PSAC, we see a clear overlap between the zone with smoke traces and the ice nucleation region. The smoke PSAC values (required in the INP estimation with Eqs. (5) and (6), Sect. 4.1) were enhanced in the humid layer down to about 9.4 km on 28 October and 8.5 km on 30 October. The smoke PSAC values were around 25 $\mu\mathrm{m}^2\,\mathrm{m}^{-3}$ at 10.4 km height on the two days (28 and 30 October) and even close to 100 $\mu\mathrm{m}^2\,\mathrm{m}^{-3}$ at the tropopause at 11 km height on 30 October 2020. The pronounced RH variability in the uppermost part of the humid layer in



Fig. 6 may indicate water vapor consumption during the ice nucleation and subsequent crystal growth processes. The smoke conversion retrieval also allows to estimate the number concentration $n_{250}$ of large particles with radius >250 nm (Eq. 2, Sect. 4) This number provides an idea about the strength of the overall INP reservoir. $n_{250}$ values of the order 6000 L$^{-1}$ correspond to PSAC values of 25 $\mu$m$^2$ m$^{-3}$. As already mentioned above, the smoke impact obviously increased from 28 to 30 October especially towards the evening of 30 October 2020 when rather strong ice nucleation was observed from 18-24 UTC

(Fig. 4).

### 5.3  Ice nucleation triggered by gravity wave activity: observation and simulation

On 1 November 2020, gravity waves (GWs) produced several wave-like cirrus features in the smoke-polluted air (9.5-11 km height). This measurement is shown in Fig. 7. It is well known that, besides the aerosol and humidity conditions, the occurrence of GWs in the upper troposphere plays a crucial role in cirrus formation processes (Kim et al., 2016; Kärcher and Podglajen,

2019; Kärcher et al., 2022). Ubiquitous mesoscale GWs generate the high cooling rates and ice supersaturation conditions required to initiate ice nucleation.

The GWs on 1 November 2020 did not cross the lidar station directly. We only saw the impact of the GWs, propagating from east to west across the Mediterranean Sea south of Limassol, on cirrus formation. GWs propagate with typical velocities of 15-20 m s$^{-1}$ and show temporal lengths of 15-25 minutes (Kalesse and Kollias, 2013). The waves on 1 November traveled almost

against the southwesterly air flow (wind direction of 230°, wind velocity of 13 m s$^{-1}$) according to the Nicosia radiosonde observations. The remaining wave-induced cirrus patterns drifted with the southwesterly winds across the lidar station. We saw the end of the wave-like cirrus features first (around 9:50 and around 11:30 UTC), and the front part of the wave-induced cirrus, i.e., the ice nucleation region, much later (around 10:30 UTC and 12:20 UTC). During the first quarter of the GW period, updrafts lofted the air parcels with vertical velocities of probably 50-100 cm s$^{-1}$. Strong cooling occurs during lofting

and generates high ice supersaturations so that heterogeneous ice nucleation can start on the available INPs. The apparent amplitude (measured with lidar when the cirrus features crossed the field site) was roughly 300-400 m on 1 November 2020. If we take ice virga development into account, the true amplitude of the gravity wave may have been around 200-300 m (see Fig. 7).

We simulated the impact of gravity-wave-induced lofting, cooling and related RH increase on ice nucleation for the atmo-

spheric meteorological and aerosol conditions measured with lidar and radiosondes on 28 and 30 October and 1 November 2020, shown in Figs. 3 and 7. The aim was to demonstrate that the smoke particles were indeed able to trigger significant ice crystal formation expressed by ice crystal number concentrations of the order of 1-100 L$^{-1}$. We considered the atmospheric conditions as measured with Nicosia radiosonde (temperature, pressure, RH profiles) and assumed that the water vapor mixing ratio in the air parcels remained constant during the lofting process. We selected the top heights of the humid layers observed

on 28 and 30 October in Fig. 6 as starting heights for the simulations of the GWs. Starting height was 9920 m on 1 November. Wave amplitudes (maximum vertical ascend or lofting length) were set to 90 m (28 October), 180 m (30 October), and 210 m (1 November). Calculation step width was $\Delta t = 20$ s (in Eqs. (5) and (6) in Sect. 4.1). This first phase lasted 300 s (full wave period of 1200 s). Thus, the average updraft velocities were 0.4 m s$^{-1}$ (28 October), 0.6 m s$^{-1}$ (30 October), and 0.7 m s$^{-1}$



(1 November). The required smoke PSAC input values were taken from the lidar observations (from Fig. 6 for the 28 and
30 October). In the computation of INP concentrations, we used the DIN and ABIFM schemes described in Sect. 4.1.

The specific environmental and ice nucleation conditions on 28 and 30 October and 1 November are depicted in Fig. 8.
The simulated GW temperature and humidity ranges ($RH_{ICE}$, relative humidity with respect to ice) for initial and full GW
periods indicate that heterogeneous ice nucleation dominated. $RH_{ICE}$ values, required for homogeneous ice nucleation, were
never reached in the simulations so that heterogeneous ice nucleation could proceed without competition with homogeneous
freezing processes. The environmental and ice nucleation conditions of the GW events mostly represent conditions where
organic aerosol (OA) particles are solid in phase when comparing with glass transition temperatures ($T_g$) of secondary organic
aerosol (SOA) particles derived from $\alpha$-pinene and naphthalene precursor gases (Charnawskas et al., 2017), ambient SOA
particles (Wang et al., 2012b), and fulvic acid particles (Wang et al., 2012b). It should be noted that even $\alpha$-pinene derived
SOA with the lowest $T_g$ does not fully deliquesce, assuming an updraft of 1 m s$^{-1}$ (Charnawskas et al., 2017), until $RH_{ICE}$
values close to and above the homogeneous freezing limit are reached. OA particles that contain molecular species with higher
molecular weights, such as fulvic acid compounds, are expected to retain the solid phase to even greater $RH_{ICE}$ values (Koop
et al., 2011). This supports the notion that smoke particles were likely solid and acted as heterogeneous INPs.

Before discussing the results of the simulations, it should be emphasized that these simplified GW simulations are performed
to provide insight in the principle ability of smoke particles to initiate cirrus formation during the passage of a GW. These
simulations do not cover more complex aerosol particle compositions and thus do not illustrate all potential complex ice
nucleation processes due to amorphous phase transitions of the various organic phases (Knopf et al., 2018). As discussed in
Berkemeier et al. (2014), the particle phase may change during the lofting of air parcels and thus the ice nucleation mode.
Here, we estimated INP number concentrations assuming immersion freezing and deposition ice nucleation. Humidification
of air during lofting leads to water uptake by the particles, causing a humidity-induced phase transition. Upon continuous
humidification the particle phase state may change from amorphous solid (glassy) via a partially deliquesced state with a solid
core residual coated by a liquid shell to a fully deliquesced liquid. This process is often kinetically limited by diffusion of
water in the particle phase so that a particle can be out of equilibrium when the timescale of humidification is shorter than
that of diffusion (Berkemeier et al., 2014; Zobrist et al., 2008). Berkemeier et al. (2014) show that, assuming updraft velocities
typical of atmospheric conditions (e.g., 0.01–10 m s$^{-1}$ and durations from 1 s to 1 h), particles can contain glassy cores even
at high RH due to slow water diffusion, as in the case for $\alpha$-pinene-derived SOA (Fig. 8) (Charnawskas et al., 2017). Based
on simulations of timescales for particle deliquescence as well as accounting for various ice nucleation pathways for a wide
variety of organic substances the authors found that, in typical atmospheric updrafts, glassy states and solid/liquid core-shell
morphologies can persist for long enough so that amorphous smoke particles could serve as INPs via deposition nucleation
and immersion freezing. Since water diffusion within OA and smoke particles, especially at these low temperatures, is not well
quantified, thereby rendering phase and morphology of the particles in response of ambient conditions uncertain, predicting
the particles' effect on cloud formation remains challenging. Hence, accurate prediction of the ice formation potential of OA
particles necessitates not only knowledge of the thermodynamic conditions under which different ice nucleation pathways
proceed and respective ice nucleation rates but also water diffusivity and particle phase state changes in response to updraft



velocity, changes in temperature and RH, and particle size and composition (Berkemeier et al., 2014; Zobrist et al., 2008;
Knopf et al., 2018; Charnawskas et al., 2017; Wang et al., 2012b; Lienhard et al., 2015)

Figures 9-11 now show the GW simulation results. In Figs. 9a-11a, the predicted ice crystal number concentration ICNC
(open symbols, ICNC=INP concentration) for each calculation $\Delta t = 20$ s as well as integral or the sum of formed ice crystals
(solid symbols) are plotted. The respective changes in ice supersaturation $S_{ICE}$ and $\Delta a_w$ (Eq. 3) are shown in Figs. 9b-11b. As
can be seen, 90-180 m lofting is sufficient to obtain ice crystal number concentrations of the order of 1-50 L$^{-1}$ via immersion
freezing (ABIFM) or 5-75 L$^{-1}$ via DIN at temperatures around $-50°$C and for the simulated PSAC values (Figs. 9a and 10a).
$S_{ICE}$ and $\Delta a_w$ increase during lofting and threshold values of 1.35-1.37 and 0.22-0.24 must be exceeded before efficient ice
nucleation is initiated by the smoke particles. Note again that $S_{ICE}$ values $\geq 1.5$ are required at temperatures around $-50°$C
before first ice crystals can be formed via homogeneous freezing. Already heterogeneously nucleated and growing ice crystals,
however, will quickly lead to a reduction of $S_{ICE}$ so that $S_{ICE} > 1.5$ may never be reached, even not during very strong updrafts
triggering strong cooling rates. In Fig. 11, similar features as in the case of the 28 and 30 October simulations are found for
the 1 November case. After exceeding a certain threshold RH, expressed in the simulation by $S_{ICE} > 1.35$, significant ice
production takes place as long as $S_{ICE}$ is higher than this threshold value, even during downward motion. INP concentrations
of the order of 15-20 L$^{-1}$ are simulated. These numbers fit well to the estimated crystal number concentrations of 6-100 Mm$^{-1}$
in Sect. 5.2.

The simulations as presented in Figs. 9-11 are sufficient to highlight the importance of GW-induced vertical motions on
ice nucleation but do not allow us to draw any conclusion on the further cirrus evolution. A sophisticated cirrus nucleation
simulation model must at least consider latent heat release due to water phase changes during ice nucleation, crystal growth
and sublimation processes, sedimentation and thus ice and humidity removal processes, and, very important, water vapor
deposition processes during the ice nucleation and ice crystal growth processes, i.e., how rapidly H$_2$O is incorporated in ice
crystals during the ice formation events and growth phases (Kärcher et al., 2022). The gas-phase diffusion of H$_2$O in air
toward ice crystals and molecular processes at the ice crystal surfaces determine the rate of irreversible vapor uptake in an ice-
supersaturated environment and thus has a strong impact on the ice supersaturation levels and the further evolution of the cirrus
system. Another aspect, that needs to be considered, is that many vertically and horizontally propagating GWs (of different
length and duration, of different propagation directions and speed) occur at the same time and lead to a random-like spectrum
of wind speed fluctuations and corresponding temperature fluctuations at a given location rather than very harmonic upward
and downward motions as simulated here (Kärcher and Podglajen, 2019; Kärcher et al., 2022). Radiative cooling at the top of
the generated cirrus layers initiate air circulation pattern that also influences vertical motions, cooling rates, ice supersaturation
levels and ice nucleation events.

## 6  Conclusion/Outlook

Based on lidar observations at Limassol, Cyprus, in the Eastern Mediterranean we found clear evidence for the impact of
wildfire smoke on cirrus formation in the tropopause region at $-47$ to $-53°$C. Optically dense smoke layers crossed the



Mediterranean Basis in October-November 2020. Several observational cases of smoke-cirrus interaction have been discussed. Simplified gravity wave simulations were in line with the observations. Lofting by 90-180 m was found to be sufficient to initiate signifcant ice nucleation on the wildfire particles which mainly contain organic material.

We will continue our research on the impact of wildfire smoke and cirrus evolution processes by analyzing lidar observations in the central Arctic in the framwork of the MOSAiC expedition in 2019-2020. We observed more than 50 cirrus systems which evolved in smoke-polluted air (Engelmann et al., 2021; Ansmann et al., 2023). The advantage of the MOSAiC campaign is that, in addition to the lidar-derived INP estimates, the ice crystal number concentration can be obtained from combined radar-lidar observations (Bühl et al., 2019) so that closure studies such as presented by Ansmann et al. (2019a) are possible. This will
increase the accuracy and reliability of the remote-sensing-based observations.

    In this article, we used a specific INP parameterization, developed for organic aerosol particles, to describe the ice-nucleating efficiency of aged wildfire smoke particles. Future work must include airborne sampling of aged smoke particles (after long range transport over weeks to months), that would allow multi-modal micro-spectroscopic analysis and chemical imaging of these aged wildfire smoke particles (Laskin et al., 2016, 2019; Knopf et al., 2014, 2022; Lata et al., 2021) coupled to
laboratory-based characterization of their ice nucleation properties.

## 7   Data availability

Polly lidar observations (level 0 data, measured signals) are in the PollyNet database (PollyNET, 2023). All the analysis products are available upon request (info@tropos.de). Radiosonde data (Nicosia-Athalassa2023) are available at https://www.meteociel.fr/observations-meteo/sondage.php?map=1 (Nicosia-Athalassa-RS, 2023) in archiving form and in a daily
basis at the WMO Information System Portal through the link https://gisc.dwd.de/wisportal/. Backward trajectory analysis has been performed by air mass transport computation with the NOAA (National Oceanic and Atmospheric Administration) HYS-PLIT (HYbrid Single-Particle Lagrangian Integrated Trajectory) model (HYSPLIT, 2023). AERONET observational data are downloaded from the respective data base (AERONET, 2023).

## 8   Author contributions

The paper was written by REM and AA under strong support of DAK. The gravity wave model was developed by AA. The data analysis was performed by KO, REM, and HB. RE, AN, AS, PS, JB, and DE were involved in the setup of the instrumentation, calibration and quality tests, and maintenance of the station. UW and DH are involved in the project EXCELSIOR. All coauthors were actively involved in the extended discussions and the elaboration of the final design of the manuscript.

## 9   Competing interests

Daniel A. Knopf is a member of the editorial board of Atmospheric Chemistry and Physics.



## 10 Financial support

The authors acknowledge the 'EXCELSIOR': ERATOSTHENES: EXcellence reseacrh Centre for Earth Surveillance and Space-Based Monitoring of the Environment H2020 Widespread Teaming project (www.excelsior2020.eu). The 'EXCELSIOR' project has received funding from the European Union's Horizon 2020 research and innovation programme under Grant
Agreement No 857510, from the Government of the Republic of Cyprus through the Directorate General for the European Programmes, Coordination and Development and the Cyprus University of Technology. The authors acknowledge support through the European Research Infrastructure for the observation of Aerosol, Clouds and Trace Gases ACTRIS under grant agreement no. 654109 and 739530 from the European Union's Horizon 2020 research and innovation programme. The PollyXT-CYP was funded by the German Federal Ministry of Education and Research (BMBF) via the PoLiCyTa project (grant no. 01LK1603A).
The study is supported by "ACCEPT" project (Prot. No: LOCALDEV-0008) co-financed by the Financial Mechanism of Norway (85%) and the Republic of Cyprus (15%) in the framework of the programming period 2014 - 2021. The lidar analysis on smoke-cirrus interaction was further supported by BMBF funding of the SCiAMO project (MOSAIC-FKZ 03F0915A). DAK acknowledges support by U.S. Department of Energy's (DOE) Atmospheric System Research (ASR) program, Office of Biological and Environmental Research (OBER) (grant no. DE-SC0021034).

*Acknowledgements.* We are very grateful to Nicosia-Athalassa radiosonde station for excellent radiosonde observations and especially to Demitris Charalambous, Meteorology Officer at the Cyprus Department of Meteorology, for the provision of historic data. We also thank the HYSPLIT team for an easy-to-use internet platform.



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



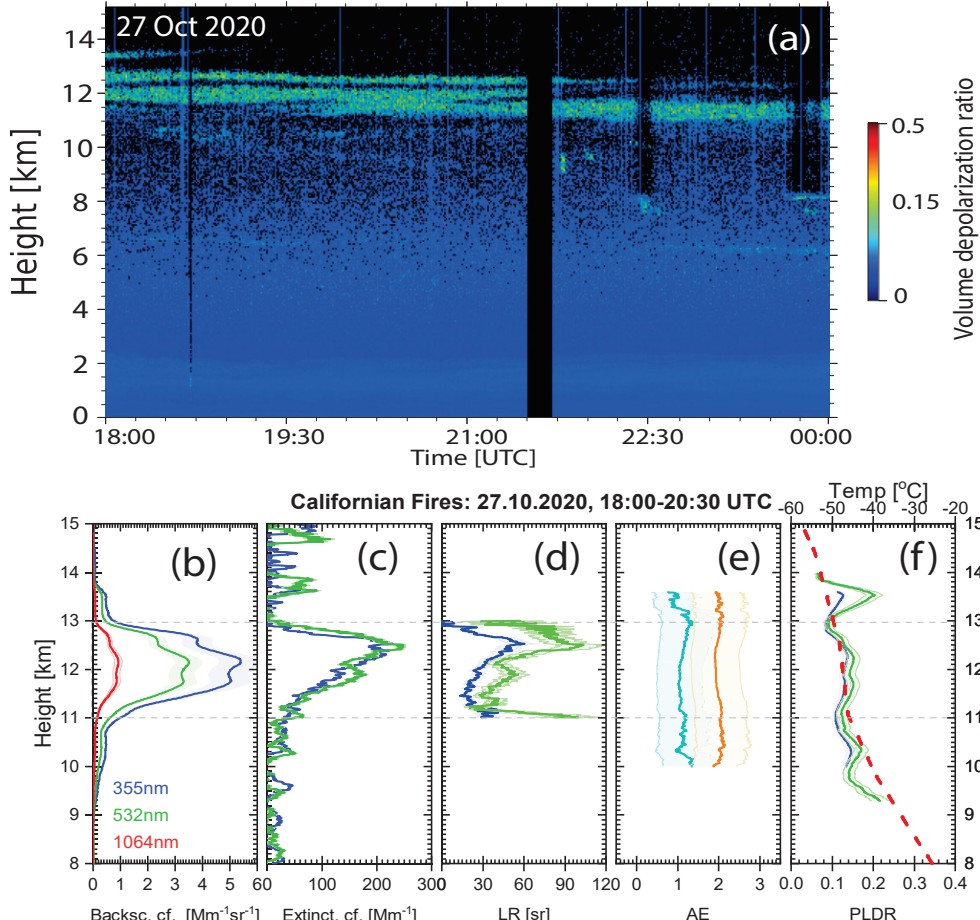

**Figure 1.** (a) Wildfire smoke layer between 11 and 13.5 km height over Limassol, Cyprus, on 27 October 2020, (b) - (f) 2.5-hour mean profiles (18:00-20:30 UTC) of particle backscatter coefficients at 355, 532, and 1064 nm, extinction coefficients and extinction-to-backscatter ratios (lidar ratios, LR) at 355 and 532 nm, backscatter Ångström exponents (AE, blue for the 355-532 nm spectrum, orange for the 532-1064 nm spectrum), and particle linear depolarization ratios (PLDR) at 355 and 532 nm. Uncertainty ranges are given, except for the extinction coefficients. Black vertical columns in (a) indicate periods with no measurements. The red dashed line in (f) shows the temperature profile with the tropopause at 11 km height.





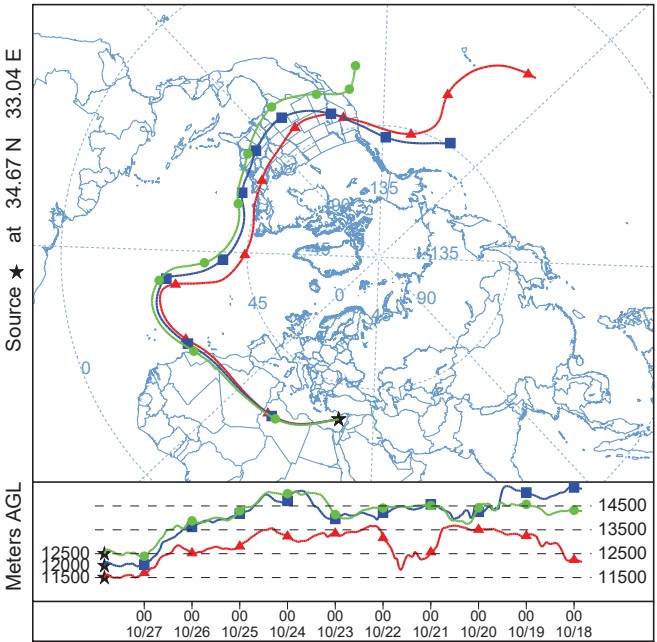

**Figure 2.** HYSPLIT 10 d backward trajectories arriving over Limassol, Cyprus (indicated by a star) on 27 October 2020, 20:00 UTC (HYSPLIT, 2023). Arrival heights are at 11500 m (red), 12000 m (blue), and 12500 m (green). The smoke source region over North America is at 12.5-14.5 km height. Smoke was probably lofted by pyroCb convection into the upper troposphere. Cloud convective processes are not considered in HYSPLIT simulations.



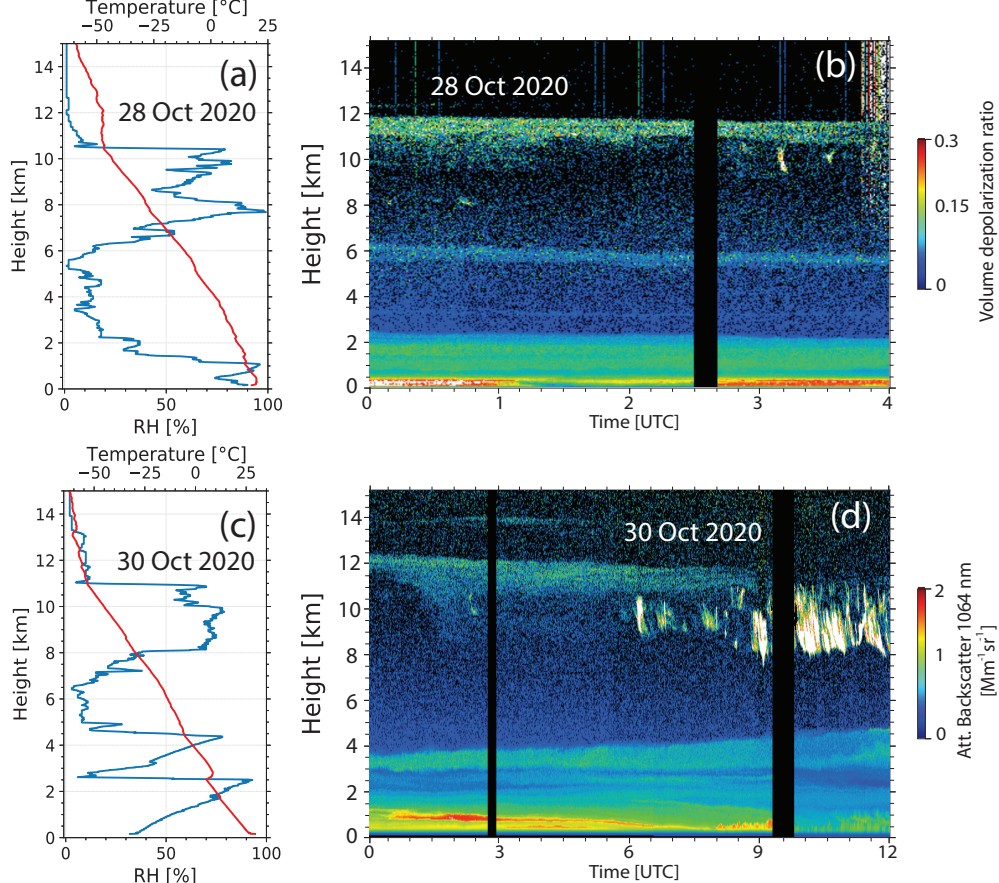

**Figure 3.** Formation of cirrus (white virga-like structures) in the lower part of an aged wildfire smoke layer (at 10-12.5 km height) on 28 October 2020 (a,b) and 30 October 2020 (c,d). In (a) and (c), radiosonde profiles of temperature (red) and relative humidity (RH, blue) are shown (radiosonde launches in (a) at 5 UTC and in (c) at 11 UTC). The sharp drop in the RH profile at 10.5 km (a) and 11km (c) indicates the tropopause. The height-time displays of the volume depolarization ratio (in b) and of the range-corrected 1064 nm backscatter signal (in d, equivalent to the 1064 nm attenuated backscatter coefficient) show smoke layers at 10-12 km and around 6 km (b) and 10-12.5 km and at 14 km height (d). Ice nucleation was most probably initiated at the top of the humid layer (close to the tropopause) at $-47°$ to $-53°$C. Black vertical columns (in b and d, 0-15 km) indicate periods with no measurements.





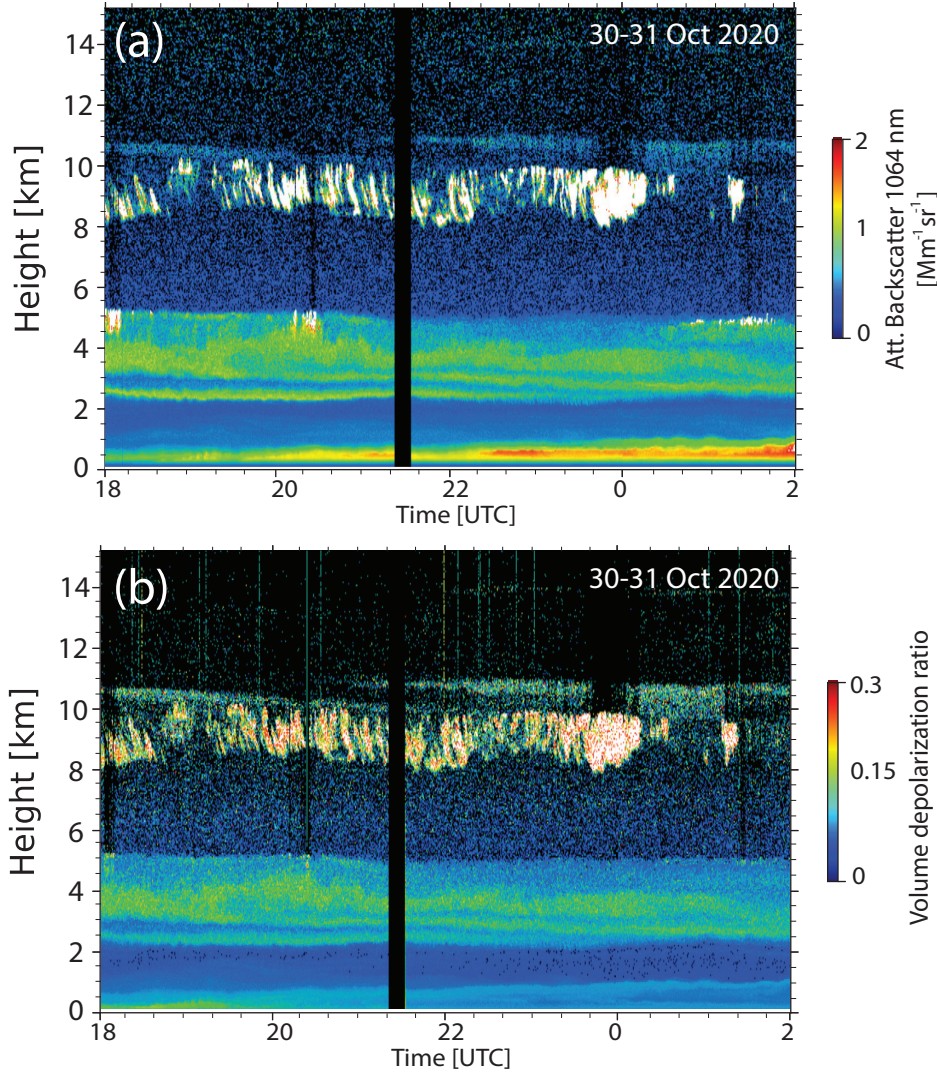

**Figure 4.** Same as Fig. 3b and d, except for 30-31 October 2020, 18:00-02:00 UTC. Ice nucleation starts at the top of the ice virga zone (white features) and thus in lower part of the smoke layers.



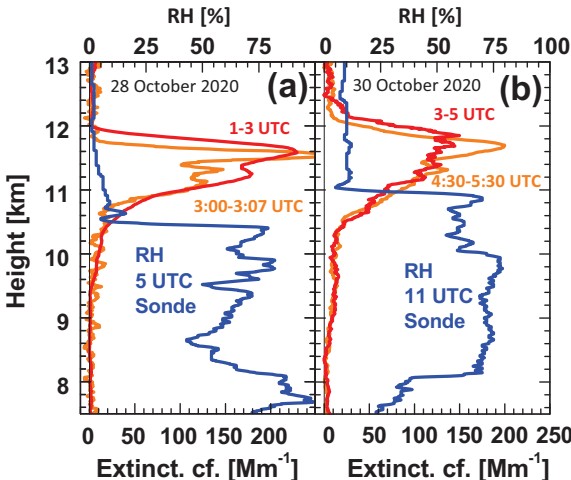

**Figure 5.** Wildfire smoke layers (orange and red particle extinction profiles) in the UTLS (orange: Klett solutions, 70 m vertical smoothing, red: Raman lidar solutions, 300 m signal smoothing) on (a) 28 and (b) 30 October 2020. The signal averaging periods are given in the panels. The 532 nm extinction coefficients are computed from the respective backscatter coefficients by multiplication with the smoke lidar ratio of 75 sr. The orange and red smoke layers partly overlap with the blue humid layers (RH radiosonde profiles, radiosonde launches at 5 and 11 UTC). The top height of the humid layer at 10.5 km (28 October) and 11 km (30 October) indicates the tropopause. Most favorable ice nucleation conditions are given just below the tropopause, i.e., in the coldest region of the troposphere.

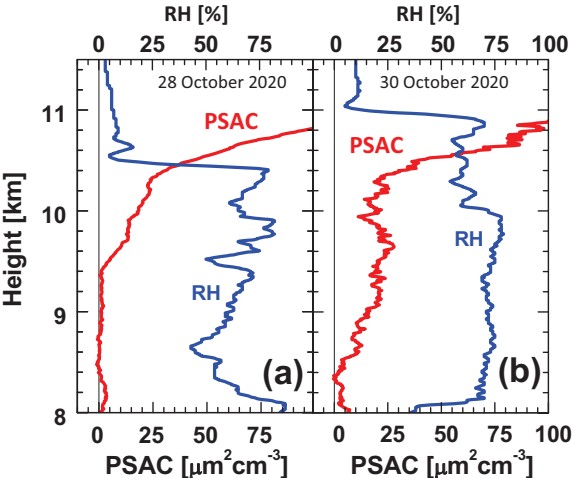

**Figure 6.** Same as Fig. 5, except for the particle surface area concentration (PSAC, red profiles). The extinction coefficients (red Raman lidar profiles) in Fig. 5 were converted into PSAC profiles by using Eq. (1) in Sect. 4. A clear overlap of the red smoke layers with the blue cirrus generation zone (RH profiles) from 9.5 km (28 October) and 8.5 km (30 October) up to the tropopause is visible.



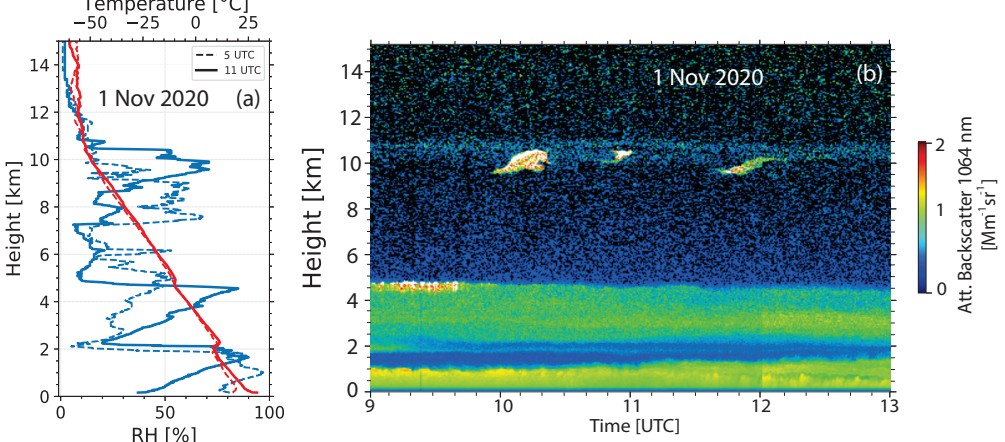

**Figure 7.** Formation of gravity-wave-induced cirrus clouds (wave-like white features around 10 km height in b) in wildfire smoke on 1 November 2020. Radiosonde profiles (launches at 5 and 11 UTC) of temperature (red) and relative humidity (RH, blue) are shown in (a). The sharp drop in the RH profile at 10.5 km (11 UTC sonde) indicates the tropopause. The height-time display of the range-corrected 1064 nm backscatter signal (denoted as attenuated backscatter) shows the smoke layer and the embedded ice clouds (in b) as well as a pronounced Saharan dust layer (2-5 km height) and the local boundary layer (up to about 1-1.5 height).



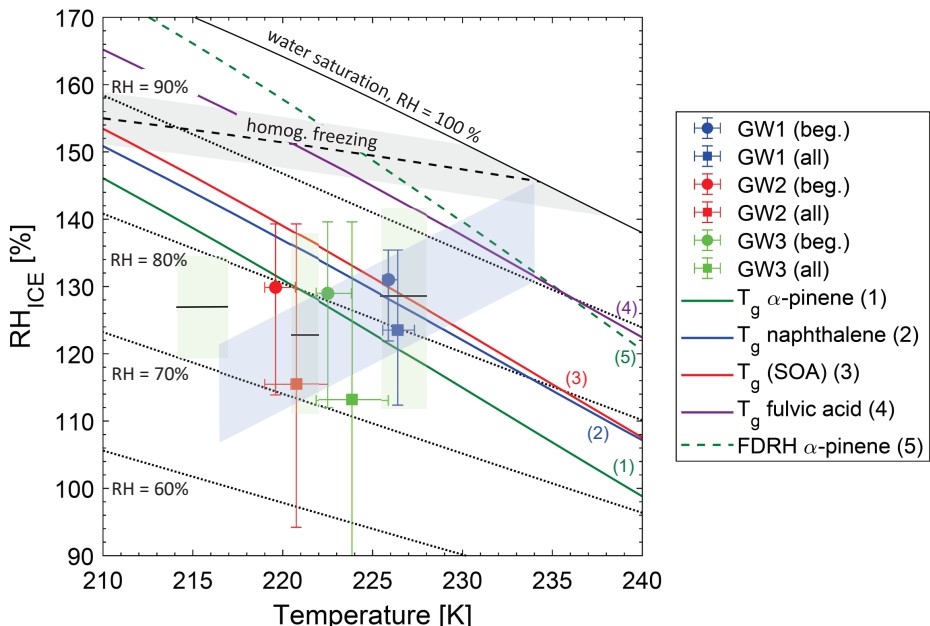

**Figure 8.** Thermodynamic conditions of observed gravity-wave (GW)-induced cirrus cloud formation (blue, red, and green symbols represent events of 28 October, 30 October, and 1 November, respectively) in comparison to homogeneous (grey area) and heterogeneous ice nucleation (greenish area), phase transitions of organic particles (colored and numbered lines), and continental cirrus conditions (bluish area). Circles represent the initial loft phase of 300 s (beg.) and squares represent the averaged values over the entire GW period (all). Solid line represents conditions of water saturation (100% RH). Dotted lines indicate constant relative humidity (90 to 60% RH from top to bottom, respectively). Dashed line and grey shading represent the homogeneous freezing limit for droplets of 10 $\mu$m in size and corresponding uncertainty (Koop et al., 2000; Koop, 2004). Greenish areas and solid lines represent the range and mean conditions of observed deposition ice nucleation by leonardite particles (Wang and Knopf, 2011) serving as surrogate of organic ice-nucleating particles. The glass transition temperature $T_g$ (molecular mobility is frozen for T<$T_g$) of laboratory generated $\alpha$-pinene SOA (green line or 1, Charnawskas et al. (2017)), naphthalene SOA (blue line or 2, Charnawskas et al. (2017)), field-derived SOA (red line or 3, (Wang et al., 2012b)), and Suwannee River Fulvic Acid particles (dark violet or 4, (Wang et al., 2012b)) are plotted. The dashed green line (or 5) displays the full deliquescence RH (FDRH) for $\alpha$-pinene SOA particles, 500 nm in diameter, for a humidification rate simulating an updraft of 1 m s$^{-1}$ (Charnawskas et al., 2017). The light blue area indicates the range of observed conditions of continental orographic wave clouds and cirrus (Heymsfield and Miloshevich, 1995).



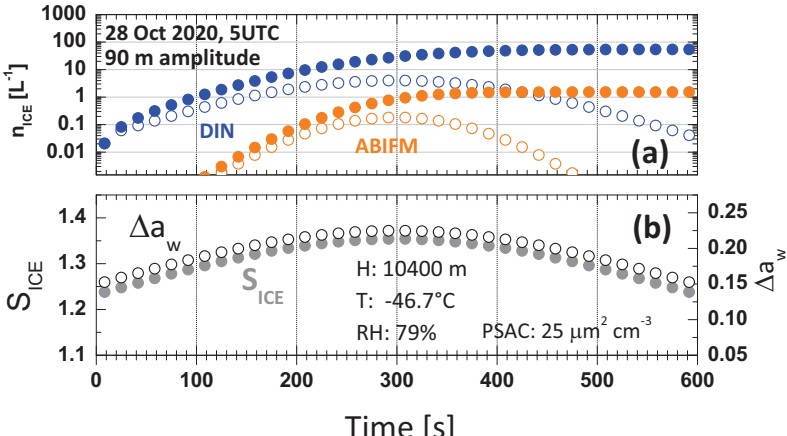

**Figure 9.** (a) Simulation of the nucleation of ice crystals ($n_{ICE}$) during the ascent phase (first 300 s) of a gravity wave (GW) with an amplitude of 120 m. Temperature (T) and RH conditions as observed with Nicosia radiosonde on 28 October 2020 (launch at 5 UTC) at the GW starting height H are given in panel (b) together with the simulated PSAC values (from the lidar observations). Immersion freezing (ABIFM) as well as deposition ice nucleation (DIN) parameterization are applied to compute $n_{INP} = n_{ICE}$ as described in Sect. 4.1. Open circles show nucleated ice crystals within each simulation step width of $\Delta t = 20$ s, and closed circles the accumulated number concentration of all freshly nucleated ice crystals. (b) Evolution of the ice supersaturation $S_{ICE}$ and the water activity criterion $\Delta a_w$ (Eq. 3) during the lofting period (0-300 s) and during the descend phase (300-600 s). The simulation belongs to the observations in Figs. 3a and b and 6a.

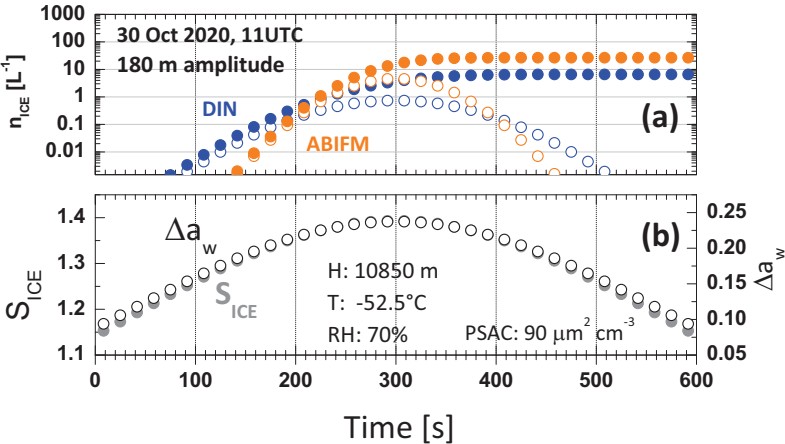

**Figure 10.** Same as Fig. 9, except for 30 October 2020 (radiosonde launch at 11 UTC) and a GW amplitude of 180 m. The simulation belongs to the observations in Figs. 3c and d and 6b.





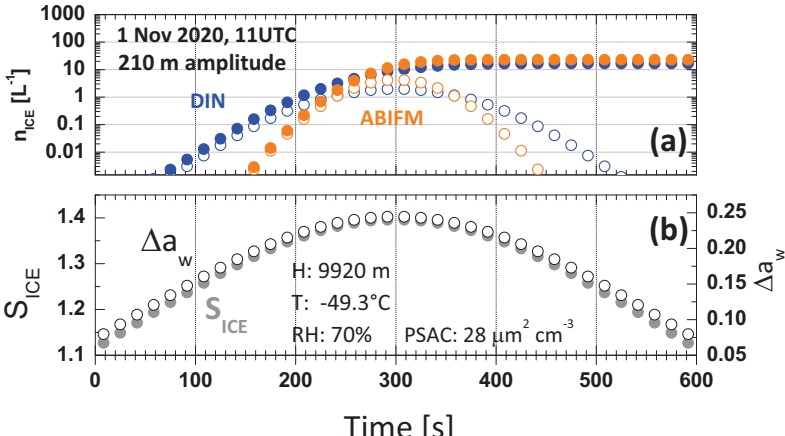

**Figure 11.** Same as Fig. 9, except for 1 November 2020 (radiosonde launch at 11 UTC). The GW amplitude is 210 m. The simulation belongs to the observations in Fig. 7.