# Peer review of "Wildfire smoke triggers cirrus formation: Lidar observations over the Eastern Mediterranean (Cyprus)"

_EGUsphere, 2023_

## Referee Comment (RC1)

Review of manuscript egusphere-2023-988, Wildfire smoke triggers cirrus formation: Lidar observations over the Eastern Mediterranean (Cyprus) for publication in ACP by Rodanthi-Elisavet et al.

The paper describes the detection of a smoke plume from forest fires in North America (California) taking 8 days to arrive in Europe in 2020 and crossing the Mediterranean from Portugal to Cyprus arriving as an aged biomass burning plume while being transported in the mid-troposphere to lower stratosphere (6-14 km). Remote sensing detection methods are used including a polarisation Raman lidar for particle backscatter and extinction coefficients as well as depolarization ratios. Cirrus formation events, virga and alternating cirrus structures from gravity waves.

The paper is of interest to the readers of ACP, in particular the cold cloud, IN and remote sensing and aerosol remote sensing community. However, the following minor revisions and not-so minor points need to be clarified. I have listed the questions/edits in order of appearance and not in order of importance.

**Line 2**: Suggest "Presently one key aspect of research is whether or not.."

**Line 17**: Delete "The" ..start sentence with "Smoke.."

**Line 27**: The authors should elaborate more on the components biomass burning particles. Here they state OA and sulfate are the major contributors but BC cores and ash or mineral particles are also known to be part of the plume. I suggest discussing their contributions here and their emission likely hood as well. Also, because later in the manuscript the authors refer to these very components (BC and minerals) as being important to determine ice nucleation and so it seems appropriate to introduce them here.

**Line 42**: Delete "up to"

**Lines 41- 47**: These are all valid claims as far as my knowledge goes, but the authors should certainly include references from the literature to support these claims that chemistry and morphology of particles change with aging and cloud processing

**Lines 54-55**: The way the sentence is structured here is awkward to me, I suggest change to "can serve as deposition ice nucleating particles (DINPs). DIN INPs is redundant.

**Line 57**: "th" should be "the"

**Line 59**: "serve as *an* INP"

**Line 60**: suggest replace "take place" with "occur"

**Line 66**: replace "efficacy" with "activity" unless a time component to nucleation is being implied here

**Line 67-69**: want not able to follow the reasoning here. The sentences above with references are support that biomass burning particles can act as INPs but then this sentence says it remains to be shown if smoke particles can influence MPC and cirrus cloud development. Perhaps the authors wish to state that the former were lab studies, and it remains to be shown in-situ is this is the case. This should be made clear. Also sentence starting with "Those INPs.." which INPs, some specificity would be good to make it more clear to the reader. At the end of this sentence, the authors can link back to the mineral/ash particles that I suggested introducing earlier, since I think the authors are referring to these particles here.

**Line 75**: Delete "here" and move "ice cloud to earlier … i.e. I suggest "In this article, we will discuss a series *of ice cloud* lidar observations that were .."
**Lines 80-83**: is this needed?

**Line 85**: inset comma after "..Raman lidar, Polly.."
**Lines 90-91**: How about marine aerosol, surely this is also part of the mix in the Eastern Mediterranean aerosol
**Line 99**: suggest replace ".Meanwhile also smoke is a topic of research (Nisantzi…" with ".. and smoke research more recently (Nisantzi.."

**Line 104**: "reflection by" should be "reflection of"
**Lines 101-105**: For a non-expert in remote sensing, this is a little too brief especially the part where the pointing to an off-zenith angle of 5° to avoid bias. Could this be elaborated a little more as it is important to distinguish the signal from ice in MPCs vs. Cirrus virga.
**Line 115**: replace "by" with "be"
**Line 118**: "signal-to-noise"?
**Lines 125-129**: I agree that $s$ is used as an input parameter for the INP parameterisation, but I don't see why the authors don't use $n_{250}$ as an input parameter for a parameterisation as well. I understand the commonly used DeMott 10 and 15 parameterisations [*Demott et al.*, 2010; *Demott et al.*, 2015] are for immersion freezing, but there are some cirrus parametrisations available for instance from the AIDA chamber work.  Is it a good assumption that all particles larger than 500 nm are available as INPs? Perhaps more explanation or justification is needed here.
**Lines 138-141**: For the assumption that the aerosol retrievals are for dry conditions, this sounds reasonable, but can the authors also state the range of RH during the cirrus free conditions for when the retrieval was conducted? That would support their assumption to neglect water uptake and depend on the dry aerosol retrievals.

**Section 4.1**:  In this section I think more justification for this method is needed or more clarification. If the authors treat the aerosol at DINPs, then why do they need to compute the INP from immersion mode at cold cirrus temperatures. The latter would only be relevant if the organic shell takes up water and dissolves, in which case if the core is BC, these would not be immersion freezing active since BC does not have active sites [*Kanji et al.*, 2020], but rather only freezes by deposition mode or PCF for temperatures below 235 K [*Chou et al.*, 2013]. And if a bulk droplet exists as these temperatures, then the freezing mechanism is homogeneous nucleation. Only when the $RH_i < 140\%$ is when PCF or DIN is considered relevant.

Also, the assumption that the particles are in equilibrium with the environment is not a good one for these conditions because the viscosity of the organic coatings really limits diffusion of water in the organics, so the very assumption of having glassy state or organic coatings, is contrary to assuming equilibrium conditions. The only relevance of immersion freezing at such cold temperatures would be if the organic coating is dissolved or diluted and the core is a mineral ash or dust compound.

In this regard, I would simplify and only consider DIN as the process and use that to retrieve INPs from the data and not immersion freezing since OA has been shown to nucleate ice via

DIN or PCF/DIN [*Kilchhofer et al.*, 2021; *Knopf et al.*, 2018; *Knopf et al.*, 2010]. And this suggestion is in line with what the authors write in section 5.1, (lines 192-196) that the fast lofting into the dry upper troposphere would limit core-shell structure formation and thus DIN would be supported over immersion freezing by water uptake.

**Lines 203-209**: The discussions here refer to supersaturated air and subsaturated air, but with respect to ice, but in Fig. 3a and c, the RH is plotted presumably with respect to water, because no SS RH regions are observable in Fig. 3a and c. Also, it is not clarified in the the caption of Fig. 3 that the RH is wrt water.

**Lines 217**: here I would reword to saying that an intensification of ice/virga was observed emerging from the smoke layer implying that strong ice nucleation by the smoke particles occurred. The way it is phrased now, is incorrect, as the process of nucleation was not observed by the remote sensing, but the ice virga evolution is observed.

**Lines 250-255**: Is there a reason why the highest number of calculated ICNC is 100 L-1 but the reservoir of INPs calculated was up to 6000 L-1, is this because not all particles are DIN in active, or the competition for water vapour? This would be better if the scale were $RH_i$ rather than $RH_w$, so the reader can tell how close to ice saturation these values are.

**Line 263 and 269**, the units provided for updraft velocity seems different here. Is that intended, if so it should be stated that the GW observed here in this work had updraft speeds much lower than typical velocities mentioned on line 263.

**Line 297**: Here the authors should add that the heterogeneous IN was likely via DIN. It does not seem plausible to me that immersion freezing is the mechanism, see comments below.

**Lines 300-315**: The assumption of immersion freezing here is flawed in my opinion or not sufficiently justified. The authors nicely explain that the shell of the aerosol or the organic phase will likely not be liquified because of the diffusion limitations of water uptake therefore the aerosol might still be highly viscous or in the glassy state, as shown in Fig 8. What then should the water uptake mechanism be, if the OA is still glassy? If water condenses onto an OA shell that is not miscible with the condensed water, then this aerosol coated with water should freeze homogeneously since the T << 235K. If the water mixes with the OA coating and freezes at these low humidities, then it can be postulated that immersion freezing is taking place with the core promoting it because the RH is below that required for homogeneous freezing of solution drops at this temperature. But it can't be that the OA is in the glassy state, and acts as a core for the water to condense and the core of the OA is initiating immersion freezing in the droplet. IF bulk water is present at these conditions, it would freeze homogeneously.

What would be the active site on the OA core promoting immersion freezing and how can this be validated given the low T where the homogeneous freezing rate of the condensed water onto the glass OA shell would be very high as well?

I agree, the data in Fig. 8 show nicely that the ice occurrence is below the glassy transition lines, so it is likely that the OA is in the glassy state, as such with the above explanations DIN

is the only likely mechanism. For immersion freezing to take place, the OA shell should become miscible with part of the water taken up.

The DIN can be readily explained, here water vapour can adsorb onto the organic shell/coating of the aerosol and eventually the adsorbed water nucleates ice, or water vapour deposited on the surface nucleates ice. One can even imagine that small cracks or pores in the organic aerosol (due to ageing while being transported) can condense small pockets of liquid water which freeze homogeneously because the temp is low enough thus inducing PCF/DIN.

What should be the reason water condenses onto a glass aerosol at sub saturated conditions, if the glassy aerosol is not absorbing water due to the high viscosity and low diffusion rates? I think these two explanations do not go hand in hand.

**Line 312**: replace "the authors" with "we"

**Line 325**: The DIN ICNC are also higher in line with this mechanism. But also what is causing the differences between the DIN assessed ICNC and the immersion freezing one?

**Line 333**: The units of ICNC is wrong
**Line 352**: "Basis" should be "Basin"

**In the conclusions** or elsewhere in the discussion, the authors should address the differences between the ICNC derived from the simulations vs. the remote sensing methods. The max for instance was 75/L vs. 100/L which uncertainties can account for his, or at least use the remote sensing derived uncertainties to say that perhaps this difference is negligible given the uncertainty in the measurement. Some acknowledgement that this are not completely similar needs to be made.

**Figures 3, 5 and 6**: I would consider changing the RH scale to $RH_i$ instead of $RH_w$. This allows evaluation of the cases of cirrus clouds based on supersaturation and the relevant phase is ice here, not liquid.

**Figure 7**. Please switch order so that the caption refers to panel a before panel b. Also the caption is disorganised, the authors refer first to panel b then to panel a and then back to b. This can be better consolidated.

**Figure 8**: The light blue area (last line caption) and the bluish area (caption line 3) are mentioned twice, but I think they refer to the same region in the plot. Please consolidate or correct. I only see one light blue/bluish area.

**References**

Chou, C., Z. A. Kanji, O. Stetzer, T. Tritscher, R. Chirico, M. F. Heringa, E. Weingartner, A. S. H. Prevot, U. Baltensperger, and U. Lohmann (2013), Effect of photochemical ageing on the ice nucleation properties of diesel and wood burning particles, *Atmospheric Chemistry and Physics*, *13*(2), 761-772, doi:10.5194/acp-13-761-2013.

DeMott, P. J., A. J. Prenni, X. Liu, S. M. Kreidenweis, M. D. Petters, C. H. Twohy, M. S. Richardson, T. Eidhammer, and D. C. Rogers (2010), Predicting global atmospheric ice nuclei distributions and their impacts on climate, *PNAS*, *107*(25), 11217-11222, doi:10.1073/pnas.0910818107.

DeMott, P. J., A. J. Prenni, G. R. McMeeking, R. C. Sullivan, M. D. Petters, Y. Tobo, M. Niemand, O. Moehler, J. R. Snider, Z. Wang, and S. M. Kreidenweis (2015), Integrating laboratory and field data to quantify the immersion freezing ice nucleation activity of mineral dust particles, *Atmospheric Chemistry and Physics*, *15*(1), 393-409.

Kanji, Z. A., A. Welti, J. C. Corbin, and A. A. Mensah (2020), Black Carbon Particles Do Not Matter for Immersion Mode Ice Nucleation, *Geophys. Res. Lett.*, *47*(11), 9, doi:10.1029/2019gl086764.

Kilchhofer, K., F. Mahrt, and Z. A. Kanji (2021), The Role of Cloud Processing for the Ice Nucleating Ability of Organic Aerosol and Coal Fly Ash Particles, *J. Geophys. Res.-Atmos.*, *126*(10), 21, doi:10.1029/2020jd033338.

Knopf, D. A., P. A. Alpert, and B. Wang (2018), The Role of Organic Aerosol in Atmospheric Ice Nucleation: A Review, *ACS Earth and Space Chemistry*, *2*(3), 168-202, doi:10.1021/acsearthspacechem.7b00120.

Knopf, D. A., B. Wang, A. Laskin, R. C. Moffet, and M. K. Gilles (2010), Heterogeneous nucleation of ice on anthropogenic organic particles collected in Mexico City, *Geophys. Res. Lett.*, *37*, L11803, doi:10.1029/2010gl043362.

---

## Referee Comment (RC2)

**Review of: 'Wildfire smoke triggers cirrus formation: Lidar observations over the Eastern Mediterranean (Cyprus)' by Mamouri et al. (https://doi.org/10.5194/egusphere-2023-988)**

**General comments**

In this paper the effects of aged smoke particles on ice nucleation are discussed. The authors study the case of 27th November to 3rd November 2020, when a smoke layer was detected at the UTLS region over Cyprus. Based on their calculated backwards trajectories the authors find that this layer originated from wild fires over North America (California). Observations of cirrus formation, virga structures and cirrus originating from gravity waves are carried out by means of active remote sensing via lidar. Additionally, simulations of the gravity waves are carried out.

The topics discussed in this paper are in the scope of ACP and the interest of its readers. The authors tackle an interesting subject and manage to successfully answer the set scientific questions. Apart from the introductory part feeling a bit segmented the manuscript is well written. The reader is introduced to the topic and the subject at hand. The methodology can easily be followed. The results and discussion are clear and well accompanied by references. Nevertheless, a list of mostly minor and technical revisions is presented in the following.

**Specific comments**

- Lines 25-27: At what altitudes did the measurements take place? Does this refer to the UTLS?
- Lines 41-47: It would be good if some papers are referenced regarding these claims.
- Lines 48-49: Are there any statistics supporting this assumption? An explanation could be added or a paper could be cited at this point supporting the claim.
- Line 65: A one-sentence explanation of the activation thresholds would be helpful before this statement.
- Line 67: 'Those INPs' refers to the minerals?
- Lines 139-141: Is there an estimation of this potential bias? Studies have shown high water supersaturations even at cirrus-free conditions. Maybe looking into the available water vapor on the measurement period at UTLS would strengthen or weaken this point especially since you have RH data from the radiosondes.
- Figure 1: The authors could specify what the uncertainty ranges are.
- Figure 1: Do the authors have an explanation to offer about the peak in PLDR at 13.5km altitude?
- Line 204: Is that the RH with respect to water or ice?
- Line 211 and Fig.3: Since the RH in Fig. 3 does not reach supersaturation I would expect that the authors are using relative humidity with respect to water (RHw). This is not necessarily wrong but would not be advisable for the study of ice crystals/cirrus. If available please use relative humidity over ice (RHi).
- Line 217: Strong ice nucleation is rather the explanation of the formation and evolution of the virga rather than an observation. Please rephrase accordingly.
- Line 240: A quantification of the good agreement would be helpful.
- Line 249: Same as Line 211, RHi would be preferable.
- Line 254: Same as Line 217
- Figure 4: Similar to above. 'Ice nucleation is expected at the top of the ice virga'.
- Figures 5, 6 & 7: Consider Using RHi instead or RHw
- Lines 268-269: How do the authors come to this conclusion?

- Line 270: Using the RHi and nucleation thresholds for the available INPs would strengthen this claim. Some INPs activate already at very low supersaturations while others need high values. Having the RHi as a reference would be beneficial.

**Technical corrections**
- Line 3: Patterns instead of pattern
- Abstract: Sentence 'Our study… to Cyprus' could be moved one sentence earlier, before 'we found… cirrus layers'. Introducing first the study before referring to results.
- Line 18: 'was transported' instead of traveled
- Line 20: 'in the future'
- Line 22: 'fire storms'
- Line 23: 'source of smoke'
- Line 30: 'with in-depth'
- Lines 30-31: 'has already been shown'
- Line 39: aged smoke particles originating from fires'
- Line 55: 'INPs' not necessary
- Line 57: 'the'
- Line 57: Here it is probably meant 'When the smoke particles take up supercooled water'
- Line 59: 'completely dissolve and become liquid (and no insoluble material within the particles is left), homogeneous freezing will take place on the resulting aqueous solutions at temperatures below −38°C'
- Line 58: warmer instead of higher temperatures to avoid potential confusion with negative values
- Line 75: Remove 'here'
- Line 75: Replace 'were generated' with 'formed'
- Lines 80-81: A new sentence for each section description. The first letter after the section number does not need to be capital
- Line 86: 'are presently'
- Line 156: Not clear what is meant. Please rephrase
- Line 178: Observations & Discussion
- Line 183: 'Figure 1 contains'
- Figure 1 legend: It would make it easier to define every panel separately
- Figure 1 legend: 'line in (f) represents the temperature'
- Line 228: remove 'of'
- Line 279: 50-100 m/s probably.
- Line 310: Does this still refer to RHice?
- Figure 8: RH could be denoted as RHw for clarity
- Figure 8: Consider changing colors of naphthalene and fulvic acid. They are not easily distinguishable
- Line 321: remove 'now'
- Line 329: Remove 'not'
- Line 331: Clarify if RHice
- Figures 9-11: Could be unified in one figure for easier intercomparison
- Line 352: 'Mediterranean basin'

---

## Author Comment (AC1)

Dear reviewer,

thank you for careful reading of the manuscript and for providing many valuable comments and ideas how to improve the paper.

A brief overview of main changes:

(1) Section 1 (Introduction) has an improved structure, is more straight forward now. Section 2 covers the instrumental part only: Sect. 2.1: CARO, Sect. 2.2: Polly, Sect. 2.3: Nicosia radiosonde. Section 3 describes the lidar data analysis, including the INP parameterizations in Sect. 3.1. We improved the DIN parameterization a bit, introduced the contact angle concept.

(2)  RH (over water) is no longer shown. In all figures, we switched to $RH_{ICE}$ .

(3) We show a new simulation figure (Fig.8) to explicitly support the gravity wave observations  on 1 November 2020. Afterwards, we show only one simulation figure (Fig.10, for 28 October) in the revised version instead of three (for 28, 30 October, 1 November) as presented in the submitted version.

(4) We went through the entire manuscript and improved the text as a whole along the comments of the reviewers.

Now the  step-by-step response to all comments with our response in blue.

The essential changes in the manuscript are indicated in BOLD.

Review of manuscript egusphere-2023-988:

The paper describes the detection of a smoke plume from forest fires in North America (California) taking 8 days to arrive in Europe in 2020 and crossing the Mediterranean from Portugal to Cyprus arriving as an aged biomass burning plume while being transported in the mid-troposphere to lower stratosphere (6-14 km). Remote sensing detection methods are used including a polarisation Raman lidar for particle backscatter and extinction coefficients as well as depolarization ratios. Cirrus formation events, virga and alternating cirrus structures from gravity waves.

The paper is of interest to the readers of ACP, in particular the cold cloud, IN and remote sensing and aerosol remote sensing community. However, the following minor revisions and not-so minor points need to be clarified. I have listed the questions/edits in order of appearance and not in order of importance.

**Line 2**: Suggest "Presently one key aspect of research is whether or not.."

**Considered!**

**Line 17**: Delete "The" ..start sentence with "Smoke.."

**Done!**

**Line 27**: The authors should elaborate more on the components biomass burning particles.

Here they state OA and sulfate are the major contributors but BC cores and ash or mineral particles are also known to be part of the plume. I suggest discussing their contributions here and their emission likely hood as well. Also, because later in the manuscript the authors refer to these very components (BC and minerals) as being important to determine ice nucleation and so it seems appropriate to introduce them here.

**We discuss all this in more detail, including dust and ash aspects (in the introduction and, later on, in the result section).**

**Line 42**: Delete "up to"

**Done!**

**Lines 41- 47**: These are all valid claims as far as my knowledge goes, but the authors should certainly include references from the literature to support these claims that chemistry and morphology of particles change with aging and cloud processing

**We rearranged the text in the introduction to meet these points. However, we try to keep the discussion short. An extended discussion (some kind of a review with references) regarding smoke transport, aging effects ,and smoke chemical and optical properties was already given in Ansmann et al. (2021).**

**Lines 54-55**: The way the sentence is structured here is awkward to me, I suggest change to "can serve as deposition ice nucleating particles (DINPs). DIN INPs is redundant.

**We changed wording (to avoid DIN INP) throughout the manuscript.**

**Line 57**: "th" should be "the"

**Improved!**

**Line 59**: "serve as *an* INP"

**Improved!**

**Line 60**: suggest replace "take place" with "occur"

**Improved!**

**Line 66**: replace "efficacy" with "activity" unless a time component to nucleation is being implied here

**Improved!**

**Line 67-69**: want not able to follow the reasoning here. The sentences above with references are support that biomass burning particles can act as INPs but then this sentence says it remains to be shown if smoke particles can influence MPC and cirrus cloud development. Perhaps the authors wish to state that the former were lab studies, and it remains to be shown in-situ is this is the case. This should be made clear. Also sentence starting with "Those INPs.." which INPs, some specificity would be good to make it more clear to the

reader. At the end of this sentence, the authors can link back to the mineral/ash particles that I suggested introducing earlier, since I think the authors are referring to these particles here.

**We changed the text and leave out confusing statements to keep the introduction short.**

**Line 75**: Delete "here" and move "ice cloud to earlier … i.e. I suggest "In this article, we will discuss a series *of ice cloud* lidar observations that were .."

**We rearranged the text here (introduction section) to be more clear and to have a better context of all the mentioned and justified points.**

**Lines 80-83**: is this needed?

**A detailed overview of the paper content (at the end of the introduction) is no longer given.**

**Line 85**: inset comma after "..Raman lidar, Polly.."

**We changed phrasing a bit…. here …**

**Lines 90-91**: How about marine aerosol, surely this is also part of the mix in the Eastern Mediterranean aerosol

**Yes, considered now!**

**Line 99**: suggest replace ".Meanwhile also smoke is a topic of research (Nisantzi…" with ".. and smoke research more recently (Nisantzi.."

**We changed the text to keep all this short.**

**Line 104**: "reflection by" should be "reflection of"

**We changed phrasing…!**

**Lines 101-105**: For a non-expert in remote sensing, this is a little too brief especially the part where the pointing to an off-zenith angle of 5° to avoid bias. Could this be elaborated a little more as it is important to distinguish the signal from ice in MPCs vs. Cirrus virga.

**We extended this part considerably (Section 2.2).**

**Line 115**: replace "by" with "be"

**Done!**

**Line 118**: "signal-to-noise"?

**Yes!**

**Lines 125-129**: I agree that $s$ is used as an input parameter for the INP parameterisation, but

I don't see why the authors don't use $n_{250}$ as an input parameter for a parameterisation as well. I understand the commonly used DeMott 10 and 15 parameterisations [*Demott et al.*, 2010; *Demott et al.*, 2015] are for immersion freezing, but there are some cirrus parametrisations available for instance from the AIDA chamber work. Is it a good assumption that all particles larger than 500 nm are available as INPs? Perhaps more explanation or justification is needed here.

**We rearranged the text a bit, but try to avoid a lengthy discussion here. We stick to our INP parameterizations (ABIFM, DIN) as given in Sect. 3.1. The primary goal of the paper is to show that smoke can lead to strong ice nucleation. In this stage of research, the ice nucleation mode is of secondary order of importance. There will be several follow-up papers (by us and by others), and then an extended discussion and the application (and testing) of a variety of INP parameterizations make sense.**

**Lines 138-141**: For the assumption that the aerosol retrievals are for dry conditions, this sounds reasonable, but can the authors also state the range of RH during the cirrus free conditions for when the retrieval was conducted? That would support their assumption to neglect water uptake and depend on the dry aerosol retrievals.

**The radiosondes show at all RH values below 60% in cloud free air, in the absence of cirrus formations over many hours. And for an RH of <60% RH there is practically no bias (at least always below 10%). We discuss that now at the end of Section 3, before the start of Sect. 3.1.**

**Section 4.1**: In this section I think more justification for this method is needed or more clarification. If the authors treat the aerosol at DINPs, then why do they need to compute the INP from immersion mode at cold cirrus temperatures. The latter would only be relevant if the organic shell takes up water and dissolves, in which case if the core is BC, these would not be immersion freezing active since BC does not have active sites [*Kanji et al.*, 2020], but rather only freezes by deposition mode or PCF for temperatures below 235 K [*Chou et al.*, 2013]. And if a bulk droplet exists as these temperatures, then the freezing mechanism is homogeneous nucleation. Only when the $RH_i$ < 140% is when PCF or DIN is considered relevant.

Also, the assumption that the particles are in equilibrium with the environment is not a good one for these conditions because the viscosity of the organic coatings really limits diffusion of water in the organics, so the very assumption of having glassy state or organic coatings, is contrary to assuming equilibrium conditions. The only relevance of immersion freezing at such cold temperatures would be if the organic coating is dissolved or diluted and the core is a mineral ash or dust compound.

In this regard, I would simplify and only consider DIN as the process and use that to retrieve INPs from the data and not immersion freezing since OA has been shown to nucleate ice via 3 , DIN or PCF/DIN [*Kilchhofer et al.*, 2021; *Knopf et al.*, 2018; *Knopf et al.*, 2010]. And this suggestion is in line with what the authors write in section 5.1, (lines 192-196) that the fast lofting into the dry upper troposphere would limit core-shell structure formation and thus DIN would be supported over immersion freezing by water uptake.

**We improved this section (4.1….. in the revised version 3.1) a bit by keeping the arguments of the reviewer in mind, but did not change much in Section 3.1. We introduced a statement of Berkemeier et al. (2014) as an argument that we present both INP parameterizations (ABIFM, DIN). The entire**

**research field is very new, and so many aspects concerning aged smoke particles and their role in ice nucleation are simply not known. Why should we already now reduce the information and application space?**

**Lines 203-209**: The discussions here refer to supersaturated air and subsaturated air, but with respect to ice, but in Fig. 3a and c, the RH is plotted presumably with respect to water, because no SS RH regions are observable in Fig. 3a and c. Also, it is not clarified in the the caption of Fig. 3 that the RH is wrt water.

**We improved all this. We clearly distinguish RH (relative humidity over water) and RH$_{ICE}$ (relative humidity over ice) now. All figures show RH$_{ICE}$ now.**

**Lines 217**: here I would reword to saying that an intensification of ice/virga was observed emerging from the smoke layer implying that strong ice nucleation by the smoke particles occurred. The way it is phrased now, is incorrect, as the process of nucleation was not observed by the remote sensing, but the ice virga evolution is observed.

**We follow this suggestion!**

**Lines 250-255**: Is there a reason why the highest number of calculated ICNC is 100 L-1 but the reservoir of INPs calculated was up to 6000 L-1, is this because not all particles are DIN in active, or the competition for water vapour? This would be better if the scale were *RH*$_i$ rather than *RH*$_w$, so the reader can tell how close to ice saturation these values are.

**We removed this confusing point (ICNC of 100 L-1 vs INP reservoir, n$_{250}$, of 6000 L-1) from the discussion. Such a discussion is not needed.**

**Line 263 and 269**, the units provided for updraft velocity seems different here. Is that intended, if so it should be stated that the GW observed here in this work had updraft speeds much lower than typical velocities mentioned on line 263.

**We removed the confusion! One of the mentioned velocities was the horizontal wind speed, i.e., the travel speed of the entire gravity wave (and not an updraft speed). This is now explicitly mentioned.**

**Line 297**: Here the authors should add that the heterogeneous IN was likely via DIN. It does not seem plausible to me that immersion freezing is the mechanism, see comments below.

**We mention at several places in the revised version that DIN was probably responsible for smoke-related ice nucleation. Only in the last figure, we show simulations for ABIFM (immersion freezing) and DIN (deposition ice nucleation). We show this just for comparison.**

**Lines 300-315**: The assumption of immersion freezing here is flawed in my opinion or not sufficiently justified. The authors nicely explain that the shell of the aerosol or the organic phase will likely not be liquified because of the diffusion limitations of water uptake therefore the aerosol might still be highly viscous or in the glassy state, as shown in Fig 8. What then should the water uptake mechanism be, if the OA is still glassy? If water condenses onto an OA shell that is not miscible with the condensed water, then this aerosol coated with water should freeze homogeneously since the T << 235K. If the water mixes with the OA coating and freezes at these low humidities, then it can be postulated that

immersion freezing is taking place with the core promoting it because the RH is below that required for homogeneous freezing of solution drops at this temperature. But it can't be that the OA is in the glassy state, and acts as a core for the water to condense and the core of the OA is initiating immersion freezing in the droplet. IF bulk water is present at these conditions, it would freeze homogeneously.

**We removed this part of the explanations in the revised version. We will present this kind of discussion (as given in the paper of Berkemeier et al., 2014) in several follow up papers. It is too much here. And we agree that DIN is most likely responsible for ice nucleation. And this message is clearly given in the paper.**

What would be the active site on the OA core promoting immersion freezing and how can this be validated given the low T where the homogeneous freezing rate of the condensed water onto the glass OA shell would be very high as well?

**As pointed out by Berkemeier, even at very low temperatures (below -50°C) but high humidities (RH above 70-90%), organic particles can become liquid if we wait long enough, at least they can develop a liquid surface ….around the glassy particle…… in that case we have immersion freezing…**

I agree, the data in Fig. 8 show nicely that the ice occurrence is below the glassy transition lines, so it is likely that the OA is in the glassy state, as such with the above explanations DIN is the only likely mechanism. For immersion freezing to take place, the OA shell should become miscible with part of the water taken up.

**OK! No problem to agree with this.**

The DIN can be readily explained, here water vapour can adsorb onto the organic shell/coating of the aerosol and eventually the adsorbed water nucleates ice, or water vapour deposited on the surface nucleates ice. One can even imagine that small cracks or pores in the organic aerosol (due to ageing while being transported) can condense small pockets of liquid water which freeze homogeneously because the temp is low enough thus inducing PCF/DIN.

**Yes, agree!**

What should be the reason water condenses onto a glass aerosol at sub saturated conditions, if the glassy aerosol is not absorbing water due to the high viscosity and low diffusion rates? I think these two explanations do not go hand in hand.

**Yes, probably! But all this remains hypothetical… and may be clarified in follow-up papers and discussions in papers, workshops, and conferences.**

**Line 312**: replace "the authors" with "we"

**The text was removed.**

**Line 325**: The DIN ICNC are also higher in line with this mechanism. But also what is causing the differences between the DIN assessed ICNC and the immersion freezing one?

**In the revised version we show ABIFM vs DIN only in the last figure. This is now a minor point of the paper. We leave out to explain why we get about 100 INPs per liter in the case of ABIFN and only 50 per liter in the case of DIN. We may do that in follow up papers.**

**Line 333**: The units of ICNC is wrong

**This part of the text is removed in the revised version**.

**Line 352**: "Basis" should be "Basin"

**Improved!**

**In the conclusions** or elsewhere in the discussion, the authors should address the differences between the ICNC derived from the simulations vs. the remote sensing methods. The max for instance was 75/L vs. 100/L which uncertainties can account for his, or at least use the remote sensing derived uncertainties to say that perhaps this difference is negligible given the uncertainty in the measurement. Some acknowledgement that this are not completely similar needs to be made.

**We do not understand the requirement here! You mean INPC, or you mean ICNC? Disregarding this special point, an uncertainty discussion in not easy in this field of observations, parameterizations, and simulations with focus on ICNC and INPC. The uncertainty for each approach is usually large (within 1-2 orders of magnitude), and not just 20-50%, if one would apply a rigorous uncertainty analysis.**

**However, the comment triggered us to provide more information about uncertainties in the INPC computations (in Sect. 3.1 and in the conclusions, Sect. 5). To our opinion, consistency checks and closure experiments are usually the only ways to obtain trustworthy conclusions for field observations by combining the observations with supporting simulations and estimations/parameterizations. All this is now briefly outlined in Sect. 3.1 and 5.**

**The goal of the paper is clear. We present observations that show a strong link between smoke layer occurrence and ice nucleation occurrence. That is the main message of the manuscript! All the simulations and estimations are given to provide some numbers in terms of INPC. And these simulated (estimated) INP number concentrations are reasonable, disregarding whether the uncertainties are within 1 order of magnitude. In follow-up papers we plan to combine estimations of INPC (from lidar observations) and estimations of ICNC (from combined lidar-radar observations) in the framework of closure studies as presented in Ansmann et al. 2019b (in the case of dust-cirrus interaction). This is planned in the case of our MOSAiC observations (Arctic smoke-cirrus observations) and Punta Arenas field campaigns (cirrus evolution in Australian wild fire smoke). But a cloud radar was not available at Limassol in October-November 2020.**

**Figures 3, 5 and 6**: I would consider changing the RH scale to $RH_i$ instead of $RH_w$. This allows evaluation of the cases of cirrus clouds based on supersaturation and the relevant phase is ice here, not liquid.

**Yes, we moved from RH (we leave that for relative humidity over water) and $RH_{ICE}$ in Figs. 3, 5, 6, and 7.**

**Figure 7**. Please switch order so that the caption refers to panel a before panel b. Also the caption is disorganised, the authors refer first to panel b then to panel a and then back to b.

This can be better consolidated.

**Improved!**

**Figure 8**: The light blue area (last line caption) and the bluish area (caption line 3) are mentioned twice, but I think they refer to the same region in the plot. Please consolidate or correct. I only see one light blue/bluish area.

**Improved! Apparent redundancy arises from the fact that a short explanation of shown curves, lines, bars, and areas is given in the first part. In the second part, the specific information and references are given to each set of curves, lines and areas.**

**References**

Chou, C., Z. A. Kanji, O. Stetzer, T. Tritscher, R. Chirico, M. F. Heringa, E. Weingartner, A. S. H. Prevot, U. Baltensperger, and U. Lohmann (2013), Effect of photochemical ageing on the ice nucleation properties of diesel and wood burning particles, *Atmospheric Chemistry and Physics*, *13*(2), 761-772, doi:10.5194/acp-13-761-2013.

DeMott, P. J., A. J. Prenni, X. Liu, S. M. Kreidenweis, M. D. Petters, C. H. Twohy, M. S. Richardson, T. Eidhammer, and D. C. Rogers (2010), Predicting global atmospheric ice nuclei distributions and their impacts on climate, *PNAS*, *107*(25), 11217-11222, doi:10.1073/pnas.0910818107.

DeMott, P. J., A. J. Prenni, G. R. McMeeking, R. C. Sullivan, M. D. Petters, Y. Tobo, M. Niemand, O. Moehler, J. R. Snider, Z. Wang, and S. M. Kreidenweis (2015), Integrating laboratory and field data to quantify the immersion freezing ice nucleation activity of mineral dust particles, *Atmospheric Chemistry and Physics*, *15*(1), 393-409.

Kanji, Z. A., A. Welti, J. C. Corbin, and A. A. Mensah (2020), Black Carbon Particles Do Not Matter for Immersion Mode Ice Nucleation, *Geophys. Res. Lett.*, *47*(11), 9, doi:10.1029/2019gl086764.

Kilchhofer, K., F. Mahrt, and Z. A. Kanji (2021), The Role of Cloud Processing for the Ice Nucleating Ability of Organic Aerosol and Coal Fly Ash Particles, *J. Geophys. Res.-Atmos.*, *126*(10), 21, doi:10.1029/2020jd033338.

Knopf, D. A., P. A. Alpert, and B. Wang (2018), The Role of Organic Aerosol in Atmospheric Ice Nucleation: A Review, *ACS Earth and Space Chemistry*, *2*(3), 168-202, doi:10.1021/acsearthspacechem.7b00120.

Knopf, D. A., B. Wang, A. Laskin, R. C. Moffet, and M. K. Gilles (2010), Heterogeneous nucleation of ice on anthropogenic organic particles collected in Mexico City, *Geophys. Res. Lett.*, *37*, L11803, doi:10.1029/2010gl043362.

**As new references we consider Chou et al. (2013), Kilchhofer et al. (2021), and DeMott et al. (2015) in the revised version.**

---

## Author Comment (AC2)

Dear reviewer,

thank you for careful reading of the manuscript and for providing many valuable comments and ideas how to improve the paper.

A brief overview of main changes:

(1) Section 1 (Introduction) has an improved structure, is more straight forward now. Section 2 covers the instrumental part only: Sect. 2.1: CARO, Sect. 2.2: Polly, Sect. 2.3: Nicosia radiosonde. Section 3 describes the lidar data analysis, including the INP parameterizations in Sect. 3.1. We improved the DIN parameterization a bit, introduced the contact angle concept.

(2)  RH (over water) is no longer shown. In all figures, we switched to $RH_{ICE}$ .

(3) We show a new simulation figure (Fig.8) to explicitly support the gravity wave observations  on 1 November 2020. Afterwards, we show only one simulation figure (Fig.10, for 28 October) in the revised version instead of three (for 28, 30 October, 1 November) as presented in the submitted version.

(4) We went through the entire manuscript and improved the text as a whole along the comments of the reviewers.

Now the  step-by-step response to all comments with our response in blue.

The essential changes in the manuscript are indicated in BOLD.

**General comments**

In this paper the effects of aged smoke particles on ice nucleation are discussed. The authors study the case of 27[th] November to 3[rd] November 2020, when a smoke layer was detected at the UTLS region over Cyprus. Based on their calculated backwards trajectories the authors find that this layer originated from wild fires over North America (California). Observations of cirrus formation, virga structures and cirrus originating from gravity waves are carried out by means of active remote sensing via lidar. Additionally, simulations of the gravity waves are carried out.

The topics discussed in this paper are in the scope of ACP and the interest of its readers. The authors tackle an interesting subject and manage to successfully answer the set scientific questions. Apart from the introductory part feeling a bit segmented the manuscript is well written. The reader is introduced to the topic and the subject at hand. The methodology can easily be followed. The results and discussion are clear and well accompanied by references. Nevertheless, a list of mostly minor and technical revisions is presented in the following.

**Specific comments**

- Lines 25-27: At what altitudes did the measurements take place? Does this refer to the UTLS?

**We are more precise now and mention: free troposphere up to the tropopause.**

- Lines 41-47: It would be good if some papers are referenced regarding these claims.

**We rearranged the text to keep the introduction as short as poosible, and with clear focus of smoke on cirrus impact.  By this rearrangement, we avoid to provide references to all these points because we did already an extended review on smoke transport, aging, and resulting changes in the physical and chemical properties with all the necessary references in Ansmann et al. (2021) and do not want to repeat all this here.**

- Lines 48-49: Are there any statistics supporting this assumption? An explanation could be added or a paper could be cited at this point supporting the claim.

**There are no statistics! There are several airborne observations (e.g., Dahlkoetter et al. 2014) that point to a core-shell structure. In addition, lidar observations (depolarization ratio) show that the smoke particles are not far away from having a perfect spherical shape. We rearranged the text to meet your comment better.**

- Line 65: A one-sentence explanation of the activation thresholds would be helpful before this statement.

**We removed the respective discussion (and our confusing hypothesis) to keep the introduction short. This discussion is not needed.**

**We mention that these minerals may cause smoke particles to be activated even at high temperature (-13°C). But we do not want to extend speculation here too much. It is just a hypothesis.**

- Line 67: 'Those INPs' refers to the minerals?

**We change the text… But yes, … minerals.**

- Lines 139-141: Is there an estimation of this potential bias? Studies have shown high water supersaturations even at cirrus-free conditions. Maybe looking into the available water vapor on the measurement period at UTLS would strengthen or weaken this point especially since you have RH data from the radiosondes.

**We improved this point after checking the radiosondes for cirrus events and cirrus free events in the upper troposphere. The uncertainty (bias) is at all less than 10% as stated in Sect.3.**

- Figure 1: The authors could specify what the uncertainty ranges are.

**Is done! We provide relative uncertainties in the text and figure captions.**

- Figure 1: Do the authors have an explanation to offer about the peak in PLDR at 13.5km altitude?

**This is the result of vertical smoothing of very noisy data! We state that at the end of Sect. 4.1!**

- Line 204: Is that the RH with respect to water or ice?

**We now introduce RH as relative humidity over water and $RH_{ICE}$ as relative humidity over ice. In the figures we now show exclusively $RH_{ICE}$ .**

- Line 211 and Fig.3: Since the RH in Fig. 3 does not reach supersaturation I would expect that the authors are using relative humidity with respect to water (RHw). This is not necessarily wrong but would not be advisable for the study of ice crystals/cirrus. If available please use relative humidity over ice (RHi).

**Agreed and Improved!**

- Line 217: Strong ice nucleation is rather the explanation of the formation and evolution of the virga rather than an observation. Please rephrase accordingly.

**Done!**

- Line 240: A quantification of the good agreement would be helpful.

**We now explicitly add the AOD values observed by AERONET. The AERONET AODs are 0.05-0.1 larger than the smoke layer AOD values. Thus, the lower troposphere (below 5 km) contributed 0.05-0.1 to the overall AOD, the rest is from the smoke layers.**

- Line 249: Same as Line 211, RHi would be preferable.

**We generally switched from RH (relative humidity over water) to $RH_{ICE}$ (relative humidity over ice) throughout the result section.**

- Line 254: Same as Line 217

**We removed the paragraph with this statement.**

- Figure 4: Similar to above. 'Ice nucleation is expected at the top of the ice virga'.

**Improved!**

- Figures 5, 6 & 7: Consider Using RHi instead or RHw.

**Done!**

- Lines 268-269: How do the authors come to this conclusion?

**To avoid a lengthy, speculative discussion (conclusion) , we removed this statement (on updraft speed).**

- Line 270: Using the RHi and nucleation thresholds for the available INPs would strengthen this claim. Some INPs activate already at very low supersaturations while others need high values. Having the RHi as a reference would be beneficial.

**We provide a simple gravity wave simulation (in a new Fig.8), directly after this statement! … with temperatures and $RH_{ICE}$ or $S_{ICE}$ values from the radiosonde on 1 November 2020. This keeps the discussion simple and explains reasonably well what supersaturation levels were needed to explain the gravity wave observation on 1 November.**

**We do not like the idea to discuss different ice activity efficiencies of particles of the same aerosol type in this paper with the main goal: lidar observations show that smoke initiated ice nucleation. Such points may be discussed in follow-up papers.**

**Technical corrections**
- Line 3: Patterns instead of pattern

**We changed the abstract text and removed pattern.**
- Abstract: Sentence 'Our study… to Cyprus' could be moved one sentence earlier, before 'we found… cirrus layers'. Introducing first the study before referring to results.

**Improved!**
- Line 18: 'was transported' instead of traveled

**Improved!**
- Line 20: 'in the future'

**Improved!**
- Line 22: 'fire storms'

**We removed this paragraph in order to have a more compact introduction.**
- Line 23: 'source of smoke'

**We removed this paragraph…**
- Line 30: 'with in-depth'

**Improved!**
- Lines 30-31: 'has already been shown'

**We removed the respective sentence to keep the introduction short.**
- Line 39: aged smoke particles originating from fires'

**We rephrased this part of the text.**
- Line 55: 'INPs' not necessary

**We rephrased all text parts with … DIN INP … accordingly….**
- Line 57: 'the'

**Improved!**
- Line 57: Here it is probably meant 'When the smoke particles take up supercooled water'

**Improved!**
- Line 59: 'completely dissolve and become liquid (and no insoluble material within the particles is left), homogeneous freezing will take place on the resulting aqueous solutions at temperatures below −38°C'

**Improved!**
- Line 58: warmer instead of higher temperatures to avoid potential confusion with negative values

**Meteorologists do not like 'warmer' temperatures! We removed the sentence!**
- Line 75: Remove 'here'

**The sentence is removed.**
- Line 75: Replace 'were generated' with 'formed'

**The sentence is gone.**
- Lines 80-81: A new sentence for each section description. The first letter after the section number does not need to be capital

**All this is gone.**
- Line 86: 'are presently'

**We changed the text.**
- Line 156: Not clear what is meant. Please rephrase

**Improved, we provide an example.**

- Line 178: Observations & Discussion

**Improved!**

- Line 183: 'Figure 1 contains'

**Improved!**

- Figure 1 legend: It would make it easier to define every panel separately

**Improved!**

- Figure 1 legend: 'line in (f) represents the temperature'

**We changed the text.**

- Line 228: remove 'of'

**Improved!**

- Line 279: 50-100 m/s probably.

**You mean line 269? However, these numbers are gone at all. We removed the statement.**

- Line 310: Does this still refer to RHice?

**We removed the respective paragraph.**

- Figure 8: RH could be denoted as RHw for clarity

**We use RH and RH$_{ICE}$ throughout the paper now.**

- Figure 8: Consider changing colors of naphthalene and fulvic acid. They are not easily distinguishable

**Improved!**

- Line 321: remove 'now'

**We changed the text.**

- Line 329: Remove 'not'

**Improved!**

- Line 331: Clarify if RHice

**Done!**

- Figures 9-11: Could be unified in one figure for easier intercomparison

**Improved! We now show just one Figure (Fig.10, for 28 October). We leave out to show very similar figures for 30 October and 1 November in the revised version.**

- Line 352: 'Mediterranean basin'

**Improved!**